# Cellular interactions within the immune microenvironment underpins resistance to cell cycle inhibition in breast cancers

Jason I. Griffiths [1,2,6] ✉, Patrick A. Cosgrove [1,6], Eric F. Medina[1], Aritro Nath [1], Jinfeng Chen[1], Frederick R. Adler [2,3], Jeffrey T. Chang [4], Qamar J. Khan[5] & Andrea H. Bild [1] ✉

Immune evasion by cancer cells involves reshaping the tumor microenvironment (TME) via communication with non-malignant cells. However, resistance-promoting interactions during treatment remain lesser known. Here we examine the composition, communication, and phenotypes of tumor-associated cells in serial biopsies from stage II and III high-risk estrogen receptor positive (ER+ ) breast cancers of patients receiving endocrine therapy (letrozole) as single agent or in combination with ribociclib, a CDK4/6-targeting cell cycle inhibitor. Single-cell RNA sequencing analyses on longitudinally collected samples show that in tumors overcoming the growth suppressive effects of ribociclib, first cancer cells upregulate cytokines and growth factors that stimulate immune-suppressive myeloid differentiation, resulting in reduced myeloid cell- CD8 + T-cell crosstalk via IL-15/18 signaling. Subsequently, tumors growing during treatment show diminished T-cell activation and recruitment. In vitro, ribociclib does not only inhibit cancer cell growth but also T cell proliferation and activation upon co-culturing. Exogenous IL-15 improves CDK4/6 inhibitor efficacy by augmenting T-cell proliferation and cancer cell killing by T cells. In summary, response to ribociclib in stage II and III high-risk ER + breast cancer depends on the composition, activation phenotypes and communication network of immune cells.

In healthy tissues, interactions among epithelial, stromal and immune cells tightly regulate cell phenotypes, proliferation and tissue composition[1]. These cellular communication networks are disrupted in tumors, with a strengthening of growth-promoting signals and weakening of growth-inhibitory controls[2–4]. The milieu of communications between cancer and non-cancer cell types can engineer the tumor microenvironment (TME) to trigger the onset of malignancy, and drive disease progression and establishment of a metastatic niche[5].

The phenotype, communication and composition of non-cancer cells within the tumor influence treatment resistance, in addition to the genetic heterogeneity and evolution of cancer cells. Diverse non-cancer cell types can modulate growth and survival signals in the TME, potentially contributing to resistance and progression. For example, tumor-associated macrophages can differentiate into an immune-suppressive M2-like phenotype instead of an immune activating M1-like state, switching signals in the TME from an anti-cancer to a pro-

[1]Department of Medical Oncology & Therapeutics Research, City of Hope National Medical Center, 1500 East Duarte Road, Duarte, CA, USA. [2]Department of Mathematics, University of Utah 155 South 1400 East, Salt Lake City, UT, USA. [3]School of Biological Sciences, University of Utah 257 South 1400 East, Salt Lake City, UT, USA. [4]Department of Integrative Biology and Pharmacology, School of Medicine, School of Biomedical Informatics, UT Health Sciences Center at Houston, Houston, TX, USA. [5]Division of Medical Oncology, Department of Internal Medicine, The University of Kansas Medical Center, Kansas City, KS, USA. [6]These authors contributed equally: Jason I. Griffiths, Patrick A. Cosgrove. ✉e-mail: jasonigriff@gmail.com; abild@coh.org

cancer state[6]. Fibroblasts can promote extracellular matrix deposition to support cancer cell growth, and endothelial cells can support angiogenesis to supply oxygen and nutrients to a growing tumor[7,8]. Corrupt cancer cell communications with these non-cancer cells of the TME allow exploitation of their regulatory functions to engineer a pro-tumor TME or avoid immune surveillance[9]. However, it remains unknown how differences in tumor communication, composition and cell phenotypes prior to and during treatment regulate tumor response to specific treatments.

Our recent studies have shown that stage II and III high-risk estrogen receptor positive (ER+) breast cancer cells upregulate growth factor receptors to amplify alternatives to estrogen growth signaling after treatment with endocrine and cell cycle therapy to bypass cell cycle arrest and promote resistance[10]. We have also shown how autocrine estrogen signaling by intrinsically resistant cancer cells can provide transferable CDK4/6 inhibitor resistance to otherwise sensitive cells in the microenvironment[11]. Targeting such aberrant communications provides therapeutic opportunities to block tumor-promoting TME interactions and control cancer proliferation[5,12].

However, CDK4/6 inhibitor mechanisms of action are clearly more diverse than originally thought[13]. Results from recent clinical trials (RIBECCA, POP and MONALEESA 2/3/7) and in vivo model studies indicate that immune interactions also play an important but complex role in determining tumor response to CDK4/6 inhibition. Beneficial treatment effects potentially include enhanced cancer immunogenicity[14], induced T cell activation[15], TME inflammation[16], and reduced abundance of regulatory T cells (Tregs) and immunosuppressive cytokines[17]. CDK4/6 inhibition also reduces T cell proliferation and induces high rates of leukopenia which is associated with shorter progression free survival[18,19]. Patients with low baseline lymphopenia and those that retain high regulatory immune populations during therapy more frequently progress[20,21]. For more patients to benefit from CDK4/6 inhibitor therapy, research is needed to determine how cancer cells of CDK4/6 inhibitor resistant tumors modify non-cancer regulatory communications to produce a supportive TME.

Communications are often mediated by ligand-receptor (L-R) interactions. Ligand signals produced by diverse cell types accumulate in the TME and bind to receptors on receiving cells. Signal transduction to the nucleus controls gene expression, culminating in changes in cell phenotype and function. Insights into how cellular phenotype influences communication between individual pairs of cells can be obtained by deciphering cell–cell interactions (CCI's) from transcriptomic data[22]. Individual level CCI's are then inferred from ligand and receptor gene expression of the sending and receiving cell and tested using permutation[23–26] or graph-based approaches[27–29]. The ability of cancer and non-cancer cell types to amass corrupting signals in the TME depends on the cellular abundance and composition of each phenotype, with rare cell types contributing sparse signals across the TME, even if individual cells are active communicators.

To understand how phenotypically diverse populations of cancer and non-cancer cells in a tumor communicate through production and receipt of signals, we apply an extended expression product method to single-cell RNA sequencing (scRNAseq) ligand and receptor transcriptomic profiles[3,22]. This extends the individual level CCI concept to measure population-level signaling received by individual cells from across all single cells profiled in a tumor (i.e., tumor-wide) or from cancer or non-cancerous populations. Accounting for both composition and phenotypic heterogeneity, using detailed annotations of the tumor's cell type composition, uncovers networks of communication between the phenotypically diverse populations of cancer and non-cancer cell types constituting the tumor. This tumor-wide perspective of communication is essential to study the cancer ecosystem as a whole. Different cell subtypes can have conflicting roles in TME engineering and the abundance and strength of signaling of each cell population influences the tumor progression. One example is the

relative abundance of the dichotomous tumor-promoting and suppressing M1/M2-like macrophage populations respectively[30].

Here, we investigate the dynamics of communication between cancer and non-cancer cell populations within early-stage ER+ breast cancer tumors during endocrine and cell cycle inhibitor treatment. We leverage serially collected patient tumor biopsies that are either growing or shrinking while on therapy, based on clinical measurements. We include both an initial discovery cohort of patients as well as a second independent cohort that effectively doubles the scale of our dataset and allows us to test and validate analyses discovered in the first cohort. The resultant collection consists of 424,581 single cells, encompassing both cancer and non-cancer cell types, obtained from 173 tumor biopsies taken from 62 patients pre, during and post treatment. By assessing the changing diversity of cell types and their communication within endocrine and cell cycle inhibitor shrinking (sensitive) and growing (resistant) tumors during treatment, we unveil ecosystem-wide variations in TME composition and cancer-immune communication. Notably, we find that CDK4/6 inhibitor resistant tumors exhibit immune suppressive cancer to myeloid signaling and hindered T cell recruitment and activation during treatment. Through in vitro coculture experiments, we discover the immune-suppressive effects of CDK4/6 inhibition and demonstrate that enhancing T cell activating communications can overcome this, reinvigorating cancer cell response to cycle therapy.

## Results
### Patient treatment, sample collection and tumor response
We studied the tumor-wide communication among cells in tumors of post-menopausal women with node positive or >2 cm ER+ and/or PR+, HER2 negative breast cancer enrolled on the FELINE clinical trial[10,31,32] (clinicaltrials.gov # NCT02712723). This trial evaluated the efficacy of combining CDK inhibition of the cell cycle with (single agent) endocrine therapy in the neoadjuvant setting. Tumor-wide communication was determined in patients randomized to receive either combined CDK inhibition and endocrine therapy (combination ribociclib = ribociclib + letrozole) ($n = 80$ patients) or endocrine therapy alone (letrozole alone = letrozole + placebo) ($n = 40$ patients). Patients were treated for six months and biopsies were collected at baseline (day 0), following treatment initiation (day 14), and end of treatment (surgery around day 180). Each patient's tumor was previously identified as growing (resistant) or shrinking (sensitive) during therapy, using multi-model tumor growth measurements over time using MRI, mammograms and ultrasounds, clinical physical examination and pathology (detailed in ref. 10). Published results show that resistant tumors exhibited regrowth during treatment, a higher proportion of tumor remaining post therapy (final size > 2/3 initial size) (t-statistic = 4.45, $p < 0.001$) and had consistent endpoint pathology response assessments (94% agreement).

### Discovery and validation cohort sequencing
The 120 patients were divided into two equally sized cohorts: a hypothesis generating discovery cohort and a validation cohort. Two-thirds of the patients in each cohort received the combination ribociclib treatment, while the remainder received letrozole alone. Single-cell RNA sequencing (scRNAseq) was performed on each serially collected sample of the tumors (detailed in ref. 10). In the discovery cohort, 35 patients provided high-quality biopsy samples yielding serial time-point scRNAseq (10X) data for analysis of cell type, phenotype, communication and composition (Fig. 1). Of these patients, 23 received combination ribociclib (13 resistant and 10 sensitive tumors) and 12 received letrozole alone (5 resistant and 7 sensitive tumors). The validation cohort was sampled and processed following the same procedures and we additionally rescued some lower quality cells to retain a greater number of non-cancer cell types (especially immune cells) across samples. From the validation cohort biopsies, high-quality

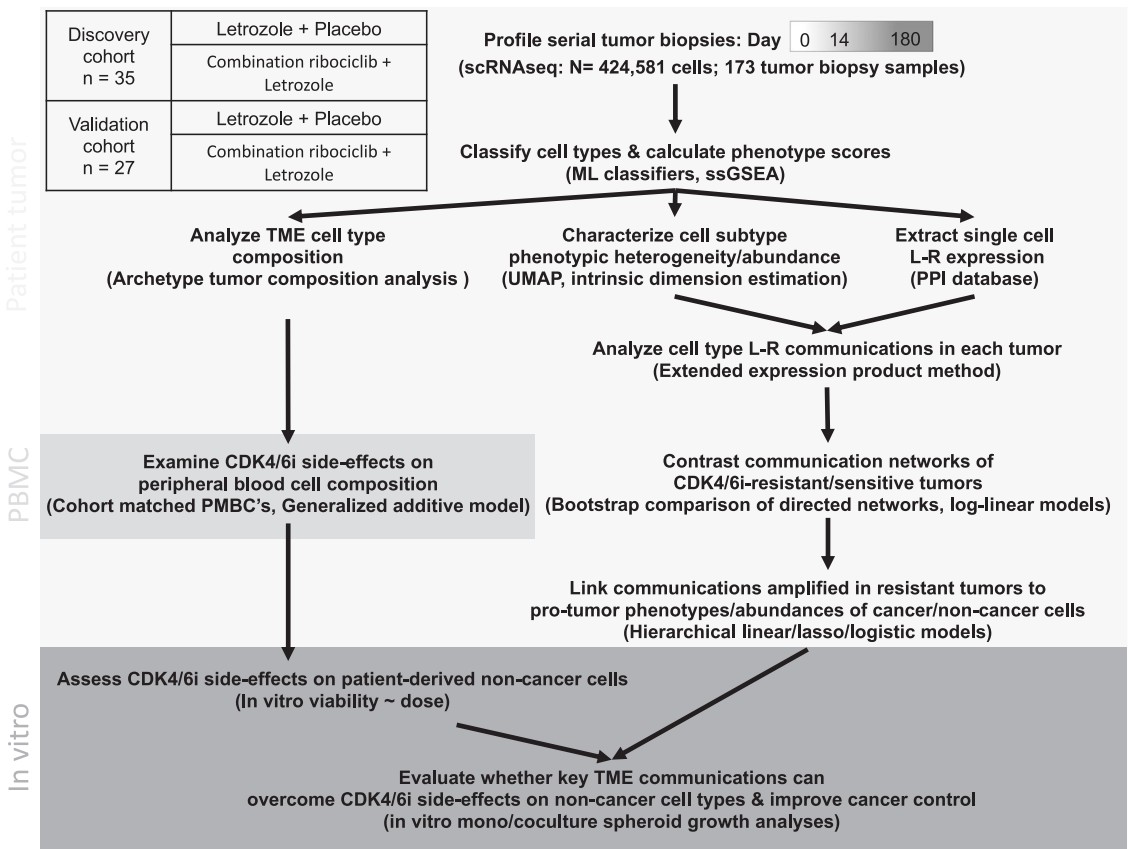

**Fig. 1 | Workflow exploring composition and communication of phenotypically diverse cancer and non-cancer cells within the tumor microenvironment of early-stage ER+ breast cancer patient tumors resistant or sensitive to CDK4/6i and endocrine therapy.** Serial single-cell RNA-seq data was generated for 62 patients by applying 10x Genomics to 173 tumor biopsy samples, collected over 3 treatment time points (Pre-treatment baseline (Day 0), Early follow-up (Day 14) and Post-treatment (Day 180)). A total of 424,581 high quality cells were transcriptionally profiled, with cancer and non-cancer cell types classified using established machine learning classifiers (see methods). Figure S1/2 show UMAP dimension reduction plots of single cell gene expression profiles, supporting cell type classification. The TME compositions of CDK4/6i-resistant/sensitive tumors were contrasted based on cell type frequencies, using pairwise distance-based dimension reduction. This identified archetypical tumor ecosystem compositions and their association with CDK4/6i response. Phenotypically diverse cell type subpopulations were resolved using cell type specific UMAP dimension reduction of ssGSEA profiles. Communication pathways through which cell subpopulations may signal were defined by 1444 ligand-receptor communication pairs with known protein-protein interaction. Networks of communication between the phenotypically diverse populations of cancer and non-cancer cell types constituting the tumor were measured, accounting for both composition and phenotype (see methods and schematic overview in Figure S3). Networks of diverse ligand-receptor communications between cancer and non-cancer cell types were compared between treatment-resistant (growing) and sensitive (shrinking) tumors. Divergent aspects of cell type communication were identified using a bootstrap comparison and verified in the independently profiled validation cohort. Specific ligand-receptor communications associated with resistance were used to predict and subsequently verify consequences on the phenotype and abundance of signal receiving cell types. The CDK4/6i treatment effects on immune cell abundance in the tumor microenvironment were then compared to temporal changes in peripheral blood mononuclear cells (PBMC) during treatment in the same patient cohort. In vitro experiments were then conducted to validate predicted effects/side-effects of CDK4/6i on cancer/non-cancer cell proliferation. Finally, we examined whether modulation of communications associated with CDK4/6i-resistance in patient tumors can overcome CDK4/6i side-effects on non-cancer cells and improve control of cancer cell growth.

serial scRNAseq data were obtained for 27 patients, of which 16 received combination ribociclib (5 resistant and 11 sensitive tumors) and 11 received letrozole alone (7 resistant and 4 sensitive tumors). The discovery and validation cohorts were sequenced independently and in subsequent analyses (below) the validation cohort was used to replicate and verify key results detected in the discovery cohort.

## Cell type annotation and verification

We obtained high quality transcriptional profiles for 424,581 single cells (41% discovery cohort, 59% validation cohort) with stringent quality controls ensuring high-coverage, low mitochondrial content, and high-confidence of doublet removal. For each cohort, broad cell types, such as epithelial cells, myeloid cells, T cells, fibroblasts, adipocytes, pericytes and endothelial cells, were discerned using singleR[33]. Cancer cells exhibited frequent and pronounced copy number amplification allowing them to be clearly identified using inferCNV[34]. Broad cell type annotations were verified by cell type specific marker gene expression and UMAP/TSNE analyses (Figure S1A)[10,35]. Granular immune subtype annotations were obtained using ImmClassifier[36]. Annotations were confirmed to be consistent between cohorts using a random forest classifier (see "methods"; Figure S1B/C).

## Shrinking tumors are immune enriched vs. cancer/stromal dominated growing tumors

Prior to examining communication between the cancer and non-cancer populations we compared the relative frequency of each cell type across tumors, allowing identification of archetypical tumor ecosystem compositions observable across early-stage ER+ breast cancers. Immune cell type abundances were found to be highly correlated with each other using hierarchical clustering. Similarly, stromal and endothelial abundances in tumors were correlated (Fig. 2A). The

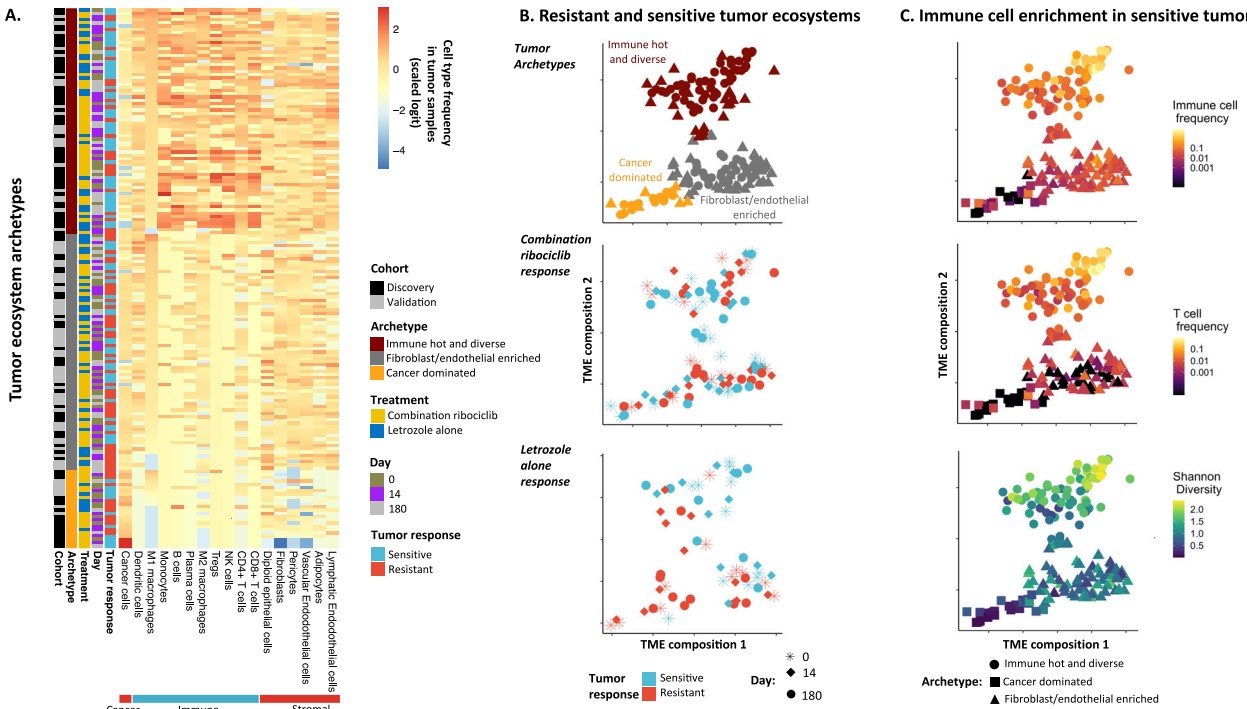

**Fig. 2 | Resistant and sensitive tumors have distinct TME compositions, with sensitive tumors having greater immune cell infiltration. A** Heatmap showing relative abundance of high-quality (HQ) cells of each cell type in tumor biopsies of the discovery and validation cohorts. Tumors samples (*y*-axis) are clustered into three compositional archetypes based on pairwise distance of compositional similarity (UMAP and Gaussian mixture model (GMM)). Cell types (*x*-axis) are clustered using hierarchical clustering to show correlation of pairwise abundance, with immune cell abundances being highly correlated with one another. Equivalent read depth cutoff applied to select HQ cells from both cohorts (see methods). **B** Distinction of three archetypal tumor compositions shown by UMAP dimension reduction using logit-Euclidean distance of compositional similarity (points = tumor samples, color = archetype, shape = cohort). Tumor data points close together have high compositional similarity. Distinct archetypal tumor compositions (top panel colors) were identified by applying the GMM to the UMAP composition space coordinates of the discovery cohort data. Then, validation cohort tumor compositions were projected into the same UMAP model space and compositional archetypes were classified by the parameterized GMM. Tumor response outcomes were associated with their compositional archetype pre, during and post treatment

with combination ribociclib (middle panel) or letrozole alone (bottom panel) (shape = timepoint). Post treatment, sensitive tumors more frequently exhibited an immune hot archetype (logistic generalized linear model predicting tumor response probability by archetype compared responsiveness of immune hot to other archetypes: Immune hot response rate relative to other archetypes: Combination ribociclib estimate = 1.7, se = 0.8, df = 29, z = 2.12, p = 0.033; Letrozole alone estimate = 3.3, se = 1.3, df = 17, z = 2.47, *p* = 0.013). **C** Tumor archetypes differ in immune cell type abundance and diversity (Shannon diversity) (color = relative abundance, shape = archetype). Comparing tumor archetypes, tumors in the immune hot and diverse archetype show increased: i) immune cell abundance (logistic generalized linear model: Increased logit immune fraction estimate = 1.47, se = 0.01, df = 166, z = 101.3, *p* = 2e-16), ii) T cell abundance (logistic generalized linear model: Increased logit T cell fraction estimate = 2.62, se = 0.04, df = 166, z = 62.4, *p* = 2e-16) and iii) Shannon diversity (ANOVA log diversity increase: estimate = 0.87, se = 0.15, df = 166, t = 5.82, *p* = 2.9e-8). Sample size = 422,635/424,581 annotated cells (ensuring equivalent HQ between cohorts) from 173 biopsy samples of 62 patient tumors at 3 timepoints. All statistical tests two-sided. Source data are provided as a Source Data file.

compositional similarity of tumor samples from the discovery cohort was determined and projected by applying the UMAP algorithm to pairwise composition distances (see methods section: Archetypal tumor compositions). Tumors with similar cell compositions clustered close together and distant from tumors with more dissimilar compositions. This analysis indicated that tumors fall into distinct archetypical ecosystem compositions associated with tumor response to treatment (Fig. 2B, C). This conclusion was supported by hierarchical clustering analyses (Figure S2A/B).

We identified three archetypal tumor compositions in the discovery cohort, using a Gaussian Mixture model and Bayesian information criterion model comparison to determine the appropriate number of TME archetypes (Figure S2C/D). Major compositional differences between archetypal compositions were identified using Dirichlet regression and correlation of TME landscape axes with cell type frequencies. The three distinct tumor archetypal ecosystems (Fig. 2B, top panel) were: i) a cancer-dominated state, ii) an immune-hot and diverse state, and iii) a fibroblast and endothelial-enriched state. Before treatment, we found growing (resistant) and shrinking (sensitive) tumors in all three states, but also a significant association of sensitive tumors with the immune-hot state (association of pre-

treatment archetype with tumor response: =11.285, df = 2, *p* < 0.005) (Fig. 2B, middle/bottom panels). After treatment, we observed a clear polarization, with 83% of shrinking treatment-sensitive tumors in the immune-hot state, and 75% of growing treatment-resistant tumors in the other two states (Figure S3).

Tumors in the validation cohort mapped into the same archetypal states in UMAP space and shrinking treatment-sensitive tumors were similarly associated with the immune-hot state. This result indicates that tumors growing during treatment become increasingly immune-cold and depauperate during treatment. Immune cell loss has been associated with resistance in various cancers[37]. ER+ breast cancers are often considered uniformly immunologically cold[38] even though multiple trials show a subset of ER+ breast cancer patient tumors respond to immunotherapy[39,40]. T cell abundances were particularly sparse in both non-immune clusters (Fig. 2C, top/middle panels), highlighting differences between tumor archetypes. In addition, the cancer cell dominant archetype showed a significantly lower Shannon diversity index score (est = −1.82, df = 165, t = −10.0, p < 0.0001) (Fig. 2C, bottom panel), and was dominated by growing treatment-resistant tumors. In summary, early-stage breast cancer tumors have three main TME compositional archetypes: Immune hot/diverse,

fibroblast/endothelial enriched or cancer dominated. Immune hot and diverse tumors, with high T cell and macrophage abundance were more sensitive to cell cycle and endocrine therapy, while immune cold tumors were more resistant.

## Deciphering communication between phenotypically diverse populations

To understand how TME composition and communication can contribute to therapy resistance, we measured tumor-wide signaling from diverse non-cancer cell sub-populations and heterogeneous cancer lineages to each receiving cell (Figure S4). Whereas cell-cell interaction approaches reveal how one cell of one type communicates with another, the extended expression-product approach measures the signal individual cells receive from across many phenotypically diverse subpopulations of cells (e.g., signal from all M1-like vs M2-like differentiated macrophages) that all contribute signals to the TME. This accounts for the abundance and ligand production of each signaling phenotype and the receptor activity of receiving cells. To do this, we dissect broad cell types into phenotypically coherent subpopulations and quantify the total contribution of signaling molecules from each group (see methods). This reveals how both phenotypic and compositional changes modify communication feedbacks and impact treatment response. We then relate these inferred LR communications back to observable changes in TME cellular phenotypes and abundances and examine in vitro how this can influence treatment response.

## Global dysregulation of communication in growing cell cycle inhibitor resistant tumors

Communication pathway scores measured the communication of ligand signals produced by one cell type population and received by individual cells of each cell type via a cognate receptor. A diverse set of 1444 ligand-receptor (LR) communication pathways were measured based on known protein-protein interactions (see methods). The overall strength of communication of one cell type with another was determined by averaging across LR communication pathway scores from each sender cell type to the receiver (Fig. 3A, B).

Across different tumors, treatments and cohorts, cancer cells contributed more communication signals to the TME than other cell types (Fig. 3A) (est = 0.67, df = 970, t = 6.26, $p < 0.0001$). In contrast, cancer cells received the least signal in general from across the TME, receiving substantially fewer communications than non-cancer epithelial, stromal and immune cells (Fig. 3B) (est = −2.45, df = 970, t = −32.0, $p < 0.0001$). However, a small subset of communications, such as growth factor communications (via ERBB family receptors) were most strongly received by cancer cells (est = 0.52, df = 967, t = 5.32, $p < 0.0001$) (Figure S5). These results indicate that cancer cells receive relatively few regulatory signals in the TME while concurrently transmitting broad and strong communications to non-cancer cells in the TME.

We next assessed how communication between cell types differed in therapy-resistant/sensitive tumors before and during treatment. Contrasting communication across many biopsies, rather than within individual biopsies, revealed the TME cell type interactions distinguishing resistant and sensitive tumors and the evolution of communication during treatment. Significant differences in communication between tumors growing and shrinking during each treatment were identified for the discovery and validation cohorts using permutation-based bootstrap randomization of the tumor response annotations for each communication pathway (see Methods) (Fig. 3C).

Prior to treatment, the resistant tumors that grew during ribociclib treatment had distinctly different communication networks from those that shrunk, showing stronger communication from cancer cells to myeloid cells (Fig. 3C, top left panel: Day 0 subpanel) (est = 0.19, z = 15.05, p < 0.0001) (Figure S6). This strengthening of cancer to myeloid cell communication in growing ribociclib-resistant tumors

was verified in the independently profiled validation cohort (est = 0.03, z = 15.23, p < 0.0001) (Fig. 3C, top right panel: Day 0 subpanel). Specific ligand-receptor (LR) communications activated in growing tumors were identified using log-linear regression with FDR multiple comparisons correction. Most activated LR communication pathways (14/20) bound to myeloid receptors known to promote an immune-suppressive myeloid phenotype (Figure S7A) (Supplementary Data 1:10). These communication pathways were not activated in growing letrozole-resistant tumors (Supplementary Data 11:20). This result revealed pre-existing communications of cancer cells with myeloid cells that may predispose resistance to cell cycle inhibition but not endocrine therapy. Activation of immune suppressive communications prior to treatment distinguished growing and shrinking (resistant/sensitive) tumors, indicating that these signals are not just indirect correlates of tumor response to treatment.

After 180 days of treatment, cell type communication diverged between tumors growing versus shrinking during therapy in both the discovery and validation cohort. In growing ribociclib-resistant tumors, all cell types developed weaker communications with cytotoxic CD8 + T cells (Fig. 3C top panels) (−0.31<est < −0.04,3< z < 13.5, $p < 0.0001$). In contrast, shrinking ribociclib-sensitive tumors retained more persistent communications with cytotoxic CD8 + T cells, with stronger signals from fibroblasts and cancer cells (Fig. 3C top right panels) (Discovery: est = 0.16, z = 13.03, $p < 0.0001$, Validation: est=0.05, z = 3.24, $p < 0.00001$) (Figure S7B). The strong fibroblast-CD8 + T cell interaction reflected increased costimulatory and recruitment integrin communications in shrinking ribociclib-sensitive tumors at day 180 (e.g., *ADAM12-ITGB1*: est = 0.83, df = 5, t = 4.77, $p < 0.05$)[41,42]. Cancer cells of ribociclib-sensitive shrinking tumors also provided greater amounts of immune-activating communications to myeloid cells, including stimulation of *CCR5/7* receptors[43] (e.g., *CCL5-CCR5*:est = 2.84, df = 6, t = 4.73, $p < 0.005$) (Supplementary Data 2). However, these communication pathways were expressed at levels too low to be verified in the validation cohort. Across cohorts, the tumor-wide decrease in CD8 + T cell communication did not occur during letrozole treatment (Fig. 3C bottom panels). In contrast, in growing letrozole-resistant tumors, cancer cells developed strong communications with stromal and epithelial cells at day 180 (Fig. 3C bottom panels: Day 180 subpanels) (Discover: 0.02< est< 0.42, 16.5< z < 31.0, $p < 0.0001$; Validation:0.004< est< 0.07, 10.2< z < 22.3, $p < 0.0001$).

The pre-treatment communication differences between growing and shrinking tumors were visualized using directed weighted network graphs of overall communication (Fig. 3D). Growing and shrinking tumors showed ecosystem-wide divergence in communication before treatment in each cohort (Fig. 3D). Growing ribociclib-resistant tumor cells exhibited stronger communication between epithelial cells and myeloid and endothelial cells whereas ribociclib-sensitive shrinking tumors had greater communication with CD8 + T cells from both cancer and non-cancer cells (Fig. 3D top panels). This was not observed in letrozole-sensitive tumors (Fig. 3D bottom panels). Overall, the dynamics of communication reveal the pre-treatment heterogeneity of cancer communication with myeloid cells that predate resistance to cell cycle inhibition but not endocrine therapy and the breakdown of tumor-regulating immune interactions in growing ribociclib-resistant tumors as well as the increase of growth-promoting interactions instead in letrozole-resistant tumors.

## Cancer cell communication with myeloid cells pre-treatment is associated with an immune-suppressing macrophage phenotype in growing ribociclib-resistant tumors

As cancer cells of growing ribociclib-resistant tumors communicated immune-suppressive signals more strongly to myeloid cells, via a range of LR pathways, we next examined the consequences on myeloid cell phenotype. Myeloid cells are a phenotypically diverse and differentiable population that sense TME conditions and regulate immune

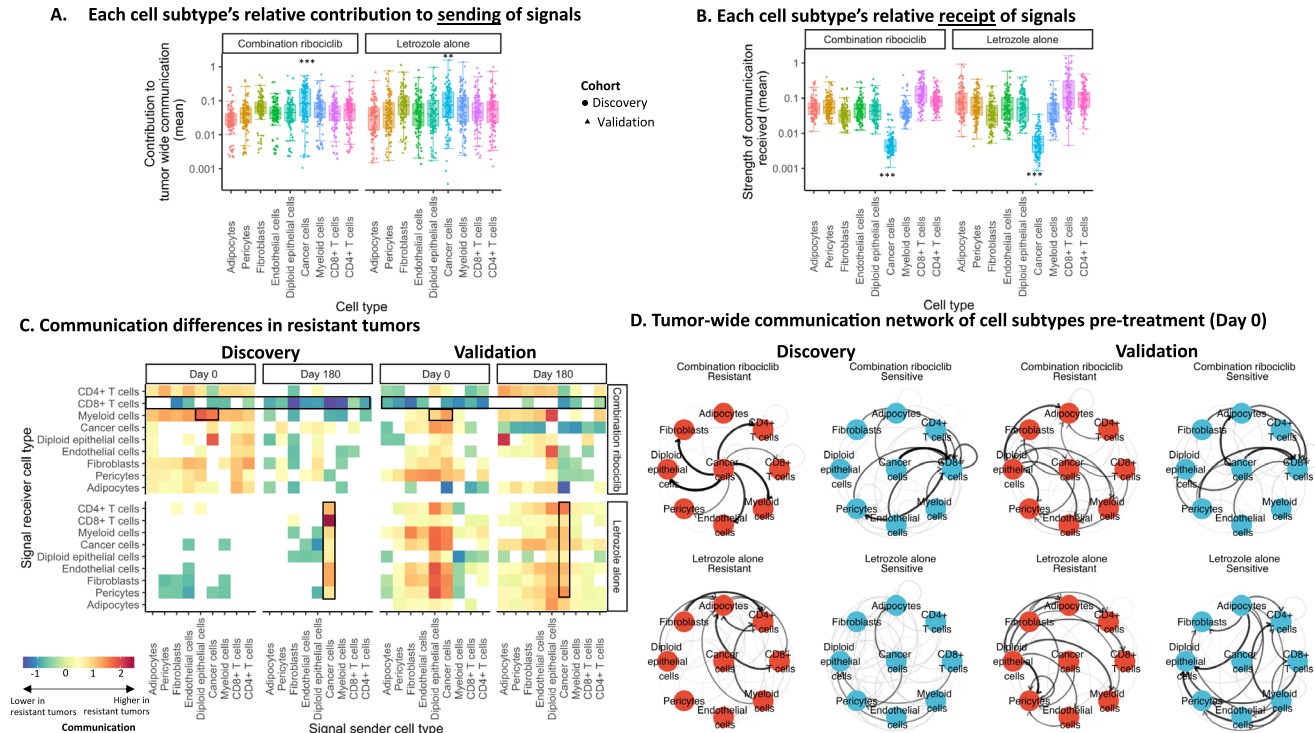

**Fig. 3 | Cell type communication differences between resistant (growing) and sensitive (shrinking) tumors pre- and post-treatment. A** Box plot showing cell type contribution to signaling with each cell type (points) across combination ribociclib or letrozole alone treated tumors of the discovery/validation cohorts (cohort = shape). Cancer cells were higher signal contributors than non-cancer cells (Linear model: Ribociclib:estimate = 0.67, se = 0.11, df = 970, t = 6.26, $p$ = 5.6e-10; Letrozole:estimate = 0.37, se = 0.13, df = 970, t = 2.89, $p$ = 0.0039). **B** Box plot showing signal strength received by each cell type (points = sender types) across tumors within treatment groups in the discovery/validation cohorts (cohort = shape). Cancer cells received fewer signals across communication pathways (Linear model: Ribociclib:estimate = −2.45, se = 0.076, df = 970, t = −32.0, $p$ = 2e-16; Letrozole:estimate = −2.47, se = 0.10, df = 970, t = −24.3, $p$ = 2e-16), despite receiving stronger growth factor signaling via ERBB receptors. Sample size for A/B = 108 cell type communication measurements for each of 9 cell types (x-axis). Measurements quantify mean signal sent/received by a cell type to/from another across resistant /sensitive tumors and sample days (0,14,180) in the discovery/validation cohort. Box elements in A/B represent median signal sent/received across cell type communication measurements (center line), upper/lower quantiles (hinges), 1.5*interquartile range (whiskers). **C** Differences in cell type communication (x-axis =

sending cell type, y-axis = receiving cells type) between tumors resistant and sensitive to combination ribociclib or letrozole alone (top/bottom subpanels) pre- and post-treatment (left/right subpanels). Cell types communicating significantly more strongly in resistant tumors (red) and sensitive tumors (blue) were identified using permutation-based bootstrap randomization tests in the discovery and validation cohorts (left/right panel). Z-scores (color intensity) quantify resistant-sensitive tumor communication differences (white = no significant difference). Black boxes indicate: i) strengthened cancer/epithelial-myeloid communication in ribociclib-resistant tumors pre-treatment, ii) reduced signaling to CD8 + T cells from across the TME in ribociclib-resistant tumors throughout treatment and iii) increased cancer signaling to diverse cell types in post-treatment letrozole-resistant tumors. **D** Network graphs showing divergence in cell type communication between resistant and sensitive tumors before treatment (top/bottom panel) in the discovery and validation cohorts (left vs right panels). Nodes represent cell types; arrow width indicates average directed communication strength across tumors. Strong communications (black) exceed 1sd above mean (other communications = gray). Sample size = 424,581 annotated cells, 1444 LR communication pathways from 173 biopsy samples of 62 patient tumors. All statistical tests two-sided. Source data are provided as a Source Data file.

responses and wound healing[7]. Signals from dead cancer cells can promote differentiation to an immune-activating M1-like macrophage or dendritic cell phenotype. However, a host of alternative signals, from cancer or non-cancer cells in the TME can promote their differentiation to an immune-suppressive M2-like phenotype that supports cancer cell proliferation and survival[6]. To assess whether the identified immune-suppressive communications from cancer cells are promoting differentiation of macrophages to a pro-tumor state, we characterized myeloid phenotypic heterogeneity by applying UMAP dimension reduction to the gene expression profile of all myeloid cells in the discovery cohort. Monocyte and dendritic cells, independently annotated using the machine learning immune classifier, formed distinct clusters and macrophages showed broad phenotypic diversity (Fig. 4A, top left panel). Assessment of genes correlated with each UMAP dimension revealed that higher UMAP2 scores characterized cells with increased M2-like macrophage polarization, with a clear expression gradient of immune-suppressive marker genes, including *CD36* (pro-fibrotic M2-like marker upregulated by *CSF1* stimulation that functions as a receptor of apoptotic cells to promote removal and

reduce inflammation), *CYP27A1* (cholesterol metabolite growth promoter), *DHRS9* (M2b Mreg marker), *LIPA* (marker of fatty acid oxidation supporting M2-like metabolism) and *PPARG* (regulator of lipid metabolism and inflammatory signaling)(Figure S8)(respective Pearson correlations = 0.49, 0.42, 0.4, 0.41, 0.36)[44–49].

Myeloid lineage differentiation was characterized using pseudotime reconstruction (see methods), revealing multiple branching differentiation trajectories that lead to the divergence between M1- and M2-like states (Figure S8A/B). Myeloid polarization was then measured by the divergence in pseudotime from the undifferentiated monocyte state (Fig. 4A, bottom left panel) and genes changing in expression with polarization characterized the transcriptional program of M1- and M2-like cells (Figure S8C). Gene set enrichment analysis showed the activation of known myeloid differentiation genes in polarized cells (Figure S9A), with established M2 markers being upregulated in cells with increased polarization and high M2-like enrichment scores (Figure S9B–D; Figure S10). Together, these analyses provided complementary lines of evidence supporting a data-driven characterization of the polarization of myeloid cells from a monocyte progenitor to

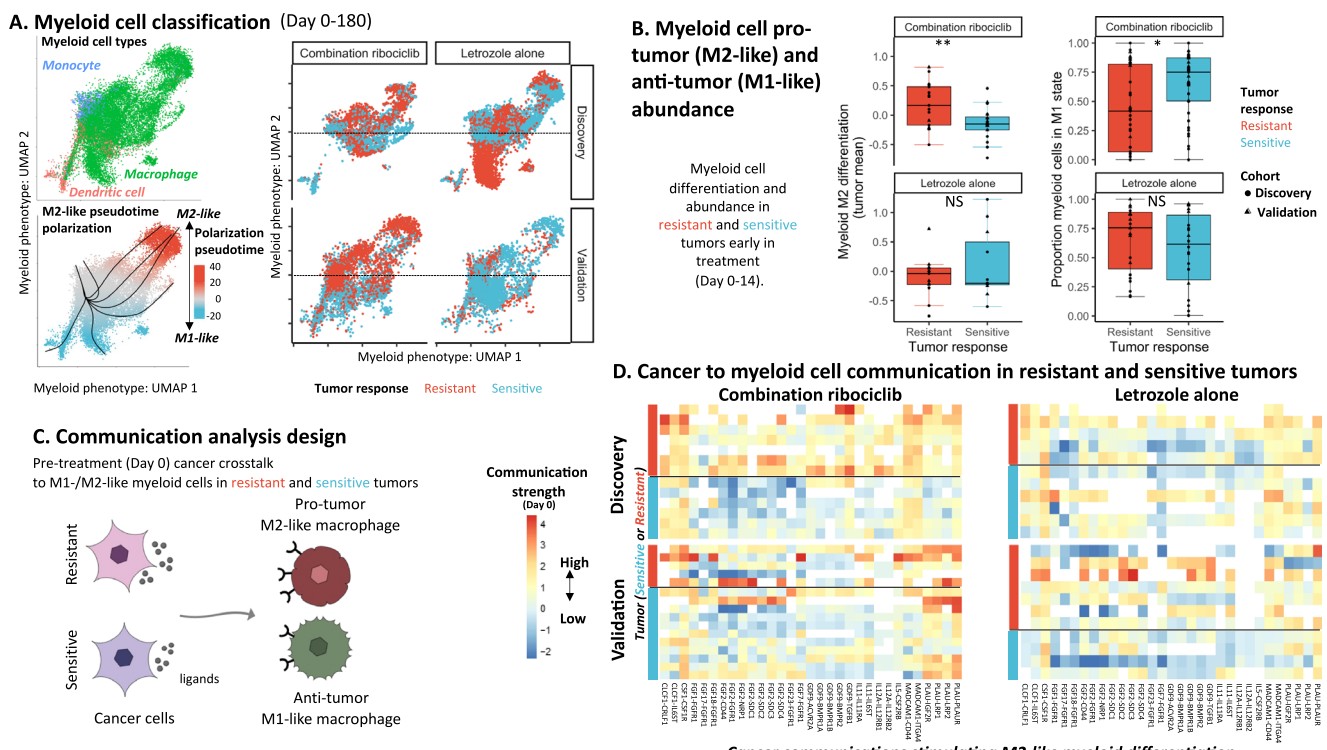

**Fig. 4 | Cancer-myeloid communications stimulate pro-tumor M2-like myeloid differentiation in CDK4/6i -resistant tumors. A** Top-left: UMAP of myeloid phenotypic heterogeneity across discovery/validation cohorts. Cells (points) with similar transcriptomic profiles clustered by ImmClassifier subtype (color). Bottom-left: Major axis of myeloid phenotypic variation reflects polarization from monocyte progenitor (gray) towards M1-like (immune-activating;blue) or M2-like (pro-tumor;red) phenotypes (polarization = pseudotime divergence;see methods). Black curves = pseudotime trajectory branching (M1/M2-like divergence). Right panel: Myeloid cell phenotypes were compared between resistant/sensitive (red/blue) tumors receiving combination ribociclib or letrozole alone in the discovery/validation cohort. Ribociclib-resistant tumors had greater M2-like differentiation (Hierarchical random effects model: Combination ribociclib:estimate = 0.337,se = 0.11, df = 32.66, t = 3.04, *p* = 0.0046;Letrozole alone:estimate = −0.30,se = 0.20, df = 20.13, t = −1.52, *p* = 0.145). **B** Left: Box plots showing increased M2-like myeloid differentiation in resistant versus sensitive tumors (red/blue) early (day 0-14) in combination ribociclib treatment (top panel)(linear model:estimate = 0.33,se = 0.11, df = 34, t = 2.99, *p* = 0.005) with no cohort-specific difference (estimate=0.058,se=0.11, df=34, t = 0.52, *p* = 0.61). Points=mean M2-like differentiation per tumor across early timepoints. No significant (NS) difference between letrozole resistant/sensitive tumors (bottom panel)(linear model:estimate = −0.23,se = 0.21, df = 21, t = 1.11, *p* = 0.28). Right: Box plots showing lower M1-like (immune-activating) myeloid cell proportion (M1/(M1 + M2)) in ribociclib-resistant tumors early in treatment (linear model:estimate = −0.24,se = 0.12, df = 64, t = −2.008, *p* = 0.048) with no cohort-specific difference (estimate = 0.21,se = 0.18, df = 64, t = 1.17,

*p* = 0.25). No difference in M1-like myeloid proportion between letrozole-resistant/ sensitive tumors (linear model:estimate = 0.14, se = 0.14, df = 37, t = 0.97, p = 0.34). Points = mean early M1-like myeloid proportion (60 tumors;multiple myeloid cell to estimate proportion). Box elements = median(center line), upper/lower quantiles(hinges),1.5*inter-quartile range(whiskers). Sample size for A/B:n = 27127 myeloid cells (Discovery:10940+Validation:16187 cells),167/173 biopsies,61/62 tumors (combination ribociclib = 37+letrozole alone = 24). **C** Schematic of cancer-myeloid communication analysis. For each tumor sample, the average strength of communication (via each LR pathway) sent by heterogeneous cancer populations to myeloid cells was measured. Significant pre-treatment communication differences between resistant/sensitive tumors were verified in the discovery/validation cohorts (log-linear regression+FDR-adjusted ANOVA). **D** Heatmap showing pre-treatment cancer-myeloid communications targeting M2-like macrophage differentiation (columns) strengthened in tumors resistant (red row annotation) versus sensitive (blue row annotation) to combination ribociclib but not letrozole alone (right-left panels) in the discovery/validation cohorts (top-bottom panels; linear model statistics in Supplementary Data 21–24). Coloration = cancer-myeloid LR-specific communication strength, showing heterogeneous pathways activated across tumors (white = no signaling detected). Sample size:268155 cancer+myeloid cells (Discovery:10940 myeloid+110568 cancer;Validation:16097 myeloid+130550 cancer),21279 genes,1444 LR pathways in 167 biopsy samples (Discovery:biopsies = 86, patients = 34;Validation:biopsies = 81, patients = 27). All statistical tests two-sided. Source data are provided as a Source Data file.

either an M1- or M2-like state being the primary axis of macrophage heterogeneity.

We next assessed differences in myeloid differentiation between treatment-resistant and sensitive tumors. Consistent with the detection of immune-suppressive cancer to myeloid communications, we found that macrophages in growing ribociclib-resistant tumors had greater M2-like differentiation prior to and throughout treatment in both the discovery and validation cohorts, while tumors shrinking during ribociclib treatment had more M1-like macrophages (Fig. 4A, right panel: left subpanels) (est = 0.34, df = 32.66, t = 3.04, *p* < 0.005). Consistent results were obtained when measuring myeloid phenotypes using knowledge-defined gene set enrichment analysis and when comparing pseudotime polarization (Figure S11; Figure S12). Antigen presenting dendritic cells were present in ribociclib-sensitive shrinking

tumors but almost entirely absent from growing ribociclib-resistant tumors. The M2-like polarization was not present in letrozole-resistant tumors (lmer est = −0.30, df = 20.13, t = −1.52, p = 0.145) (Fig. 4A, right panel: right subpanels). One growing letrozole-resistant tumor of the discovery cohort had particularly strongly M1-like macrophages. These cells had an unusual myeloid phenotype, with considerable *ERBB4* growth factor receptor upregulation which is associated with *NRG4* mediated apoptosis of pro-inflammatory macrophage[50]. We repeated analyses with this patient's myeloid cells excluded to verify our conclusions.

To determine whether the pre-existing M2-like macrophage polarization was clinically predictive of tumor response, we analyzed average myeloid phenotypes of tumors early in treatment (day 0-14) in the independent discovery and validation cohorts. We verified that

growing ribociclib-resistant tumors had more M2-like myeloid cells early in treatment compared to ribociclib-sensitive tumors shrinking during treatment (est = 0.33, df=34, t = 2.99, $p < 0.005$) (Fig. 4B, left panels). This pattern did not differ between the discovery and validation cohort (est = 0.058, df = 34, t = 0.52, $p = 0.61$). Across the discovery and validation cohorts, ribociclib-sensitive shrinking tumors instead had a higher proportion of their myeloid cells in the immune-activating M1-like state early in treatment (est = 0.24, df = 64, z = 2.00, $p < 0.05$) (Fig. 4B, right panels). The balance of immune-activating versus immune-suppressive myeloid cell frequency was more indicative of response than the total myeloid abundance.

We next measured the between tumor heterogeneity in the cancer cell communications promoting macrophage polarization and tumor growth during ribociclib treatment. We first verified the reliability of M2-like polarizing communication measurements through comparison of known M2-like differentiation communication pathways, including *CSF1-CSF1R*, *ADAM10-AXL* and *ZP3-MERTK*, to M1-/M2-like cells (Figure S13A)[51,52]. We then performed a supervised analysis of the strength of cancer to macrophage signaling across tumors in the discover and validation cohorts (Fig. 4C) (Supplementary Data 21–24). Using the discovery cohort, we identified the range of cancer ligands used to modulate macrophage phenotype and associated these with tumor response. We selected communication pathways through which cancer cells: a) signal more strongly with macrophages in growing than shrinking tumors or b) have stronger inferred cell-cell interactions with M2-like versus M1-like macrophages. We then compared the strength of each M2-like stimulating communication from cancer to myeloid cells in growing (resistant) and shrinking (sensitive) tumor samples taken prior to treatment (Day 0) in each cohort (Fig. 4D). Cancer cells of growing ribociclib-resistant tumors used a diverse set of M2-like differentiation stimulating communication pathways as shown by a range of markers: *CSF1*, *CLCF1*, several *FGFs* (1/2/7/17/18/23), the *TGF* family member *GDF9*, interleukin 5/11/12, *MADCAM1* and *PLAU* (Fig. 4D, left panel) (Supplementary Data 21–22). These ligands have been established to contribute to macrophage M2-like polarization and immune suppression via stimulation of macrophage receptors *CSF1R*, *CSF2RB*, *NRP1* and *IL6R*[53–59]. In contrast, macrophages in growing letrozole-resistant tumors did not consistently receive these M2-like differentiation communications from cancer cells (Fig. 4D, right panel) (Supplementary Data 23–24). We verified that the cancer cells were the primary contributors of these M2-like differentiation communications in growing ribociclib-resistant tumors by comparing the signaling contribution of each non-cancer and cancer cell type (Figure S13B). In these tumors, the cancer cell contribution was 35% greater than the total signal from across all cell types within the shrinking ribociclib-sensitive tumors. The ability of cancer cells to facilitate polarization of monocytes toward an M2-like phenotype has been previously characterized in vitro under coculture experiments across multiple cancer types including glioblastoma, lung, and breast cancers[60–62]. We confirmed using in vitro cancer-myeloid cocultures that breast cancer cell communications can induce the predicted form of M2-like myeloid differentiation. We compared the transcriptomic profiles of monocytes grown alone versus when cocultured with breast cancer cells in a transwell setting to prevent direct contact. Differential expression analysis showed the increased myeloid expression of many established M2-like marker genes when cocultured with cancer cells (Figure S14). This experiment validated the ability of breast cancer cells to induce myeloid polarization towards an immune-suppressing M2-like phenotype.

A comparison of the strength of each M2-like differentiation communication across tumors showed that the heterogeneous cancer populations of each growing ribociclib-resistant tumor used unique combinations of these M2-like differentiation communications (Supplementary Data 21–22). Additional M2-like differentiation communications not identified in the discovery cohort were detected in the validation cohort's growing ribociclib-resistant tumors. This diversity

suggests that directly blocking all M2-like differentiation signals would be challenging. Together these results reveal that M2-like macrophage polarization was likely driven by heterogeneous cancer communications from cancer cells that evolved prior to treatment.

## Growing tumors enriched in immune suppressing myeloid cells exhibit diminished interleukin signaling and reduced CD8 + T cell recruitment and activation during ribociclib treatment

Macrophage polarization to an M1-like or M2-like phenotype is expected to drive either anti-tumor immune activation or pro-tumor immune suppression, respectively. We therefore examined the communications of macrophages with cytotoxic CD8 + T cells across treatment resistant and sensitive tumors. We first tested the reliability of macrophage to CD8 + T cells communication measurements by testing known signaling effects of these cells and confirmed that individual M1-like macrophages sent stronger immune-activating signals (e.g., *CXCL9*) and M2-like macrophages sent stronger immune-suppressing signals (e.g., *CXCL13* and *CD47*) (Figure S15A)[63,64].

We next determined how the M2-like polarization of the entire myeloid cell population in tumors growing during ribociclib treatment impacted the communication of immune-activating signals to T cells with an analysis overview presented in Fig. 5A. Our analysis identified specific macrophage to T cell inflammatory cytokine communications that diverged during treatment between ribociclib-resistant /sensitive tumors in the discovery cohort. These were then assessed in the independent validation cohort. Hierarchical random effects models quantified macrophage to T cell communication differences at end of treatment between growing and shrinking tumors of the discovery cohort, while controlling for background patient specific variation in immune states. This analysis revealed that CD8 + T cells of growing ribociclib-resistant tumors received fewer interleukin 15 and 18 (*IL-15*, *IL-18*) activation signals from macrophages compared to shrinking ribociclib-sensitive tumors; with less stimulation of interleukin receptors 2, 15 and 18 on T cells (Fig. 5B top left) (*IL-15-IL2RA*: est = 0.035, df = 180.4, t = 3.85, $p = 1.6e-4$; *IL-15-IL-15RA*: est = 0.026, df = 132.8, t = 2.72, $p = 7.3e-3$; *IL-18-IL-18R1*: est = 0.04, df = 197.67, t = 4.92, $p = 1.8e-6$)(Supplementary Data 25). These receptors are essential for survival, proliferation, and effector differentiation respectively[65–68]. Using the independently profiled validation cohort, we verified that T cells of growing ribociclib-resistant tumors received fewer of each of these IL-15/18 activation signals during treatment, while these communications increased in shrinking ribociclib-sensitive tumors (Fig. 5B top right) (Supplementary Data 26–29). This lack of T cell activation by myeloid cells was not observed in growing letrozole-resistant tumors of the discovery or validation cohort (Fig. 5B bottom).

We compared this tumor-wide communication from across macrophages with the ability of individual M1-like macrophages to crosstalk with CD8 + T cells. We found that significantly fewer T cell activating communications were sent per M1-like macrophage in growing ribociclib-resistant tumors, indicating the suppressed activity of cells in this key immune stimulating population (Figure S15B). We measured the overall immune activating myeloid to CD8 + T cells communication across immune activating inflammatory cytokine pathways (identified using the gene-ontology database[69]). Hierarchical regression analysis of this data also indicated that the M2-like dominated macrophage populations of growing ribociclib-resistant tumors provided progressively fewer immune activating signals to CD8 + T cells throughout treatment (est = −0.023, df = 778.8, t = −15.18, $p < 0.0001$), whilst immune activation was maintained in shrinking ribociclib-sensitive tumors (stronger end of treatment communication vs growing ribociclib-resistant tumors: est = 0.024, df = 778.8, t = 6.79, $p < 0.0001$) (Figure S15C).

Phenotypic activation of CD8 + T cells of the discovery and validation cohort was measured using a CD8 T cell specific ssGSEA

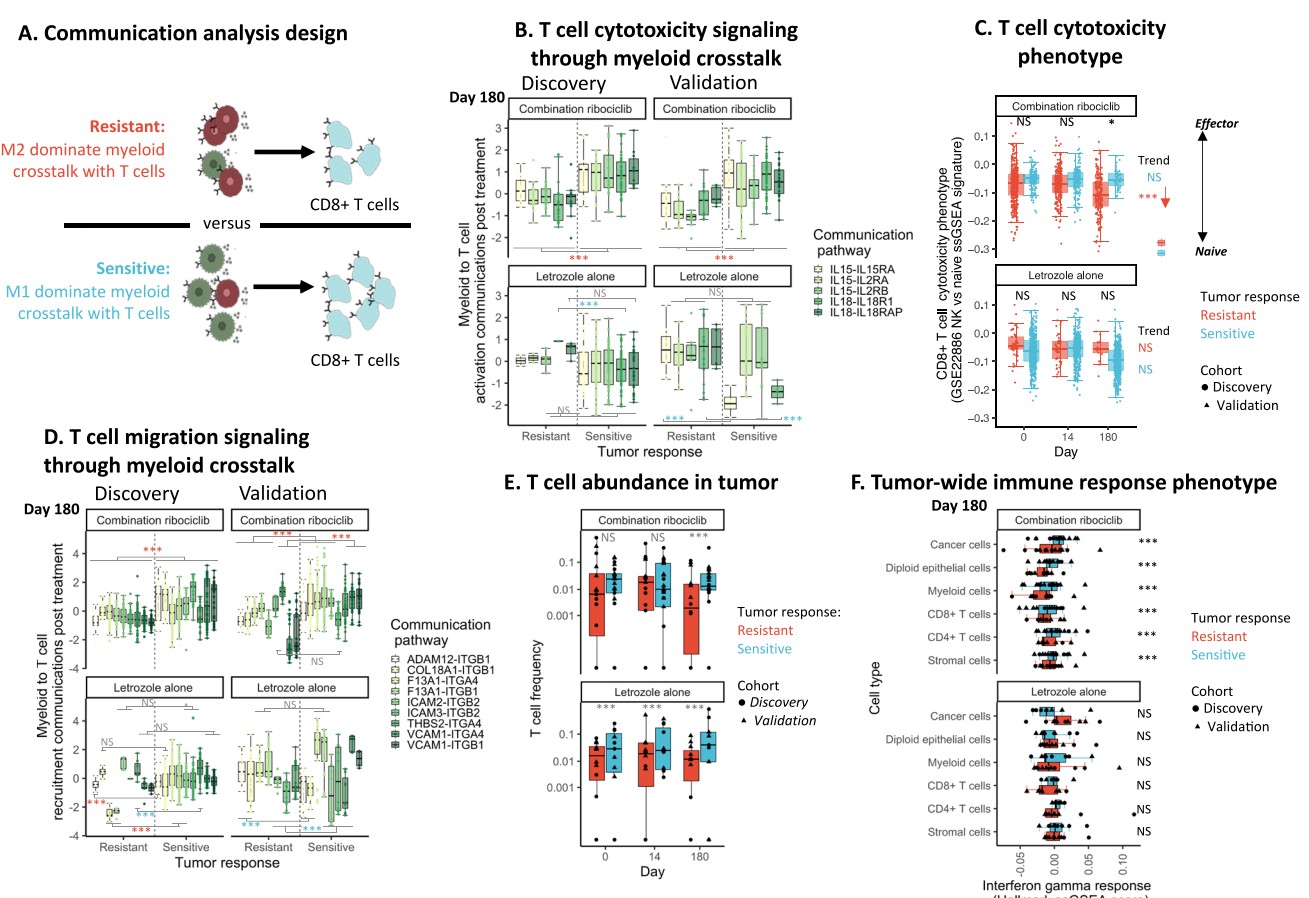

Fig. 5 | **Anti-tumor myeloid communications maintain CD8 + T cell cytotoxicity and migration in CDK4/6i-sensitive tumors. A** Schematic: communication pathway analysis revealing consequences of M2-like polarization of myeloid populations on immune-activating signals to CD8 + T cells. Cytokine communication strengths from phenotypically diverse myeloid populations to CD8 + T cells were contrasted between resistant/sensitive tumors throughout treatment (Hierarchical random effects models (HRE)+Satterthwaite t-test) and verified in discovery/validation cohorts. **B** Box plot: reduced CD8 + T cell activating communications from myeloid populations in ribociclib-resistant versus sensitive tumors (top panels) post-treatment (Day 180) in the discovery/validation cohorts but not under letrozole alone (NS = not significant)(HRE statistics in Supplementary Data 29)(color = Interleukin communication). Sample size: 42049 cells (Myeloid:Discovery = 10940+Validation = 16097; T-cell:Discovery = 3496+Validation = 11516), 139 cytokine receptors (from gene-ontology), 347 LR communications, 134 tumor samples (Discovery:biopsies = 59, patients = 28,Validation:biopsies = 75, patients = 27), 3 timepoints. **C** Box plot: reduced CD8 + T cell effector differentiation in ribociclib-resistant tumors during treatment (top) causing lower effector function post-treatment versus sensitive tumors (HRE::differentiation trend:estimate = −1.33,se = 3.13e-5, df = 112.7, t = −4.26, p = 2.2e-5;post-treatment:estimate=1.63,se=7.7e-5, df=72.3, t = 2.11, p = 0.035). No trend in differentiation under letrozole alone (bottom)(estimate = −1.29,se = 1.35e-4, df = 141.7, t = −0.94, p = 0.35). Effector differentiation of discovery/validation cohort (shape) measured using CD8 + T cell specific ssGSEA pathway contrasting naïve/cancer-killing effector expression (GSE_22886_Naive_CD8_T_cell_vs_NK_cell_up). Sample size: 2579 CD8 + T-cells (Discovery = 1977+Validation = 602 cells), 116 tumor samples (Discovery:biopsies = 56, patients = 28; Validation:biopsies = 60, patients = 26), 3 timepoints. **D** Box plot: reduced CD8 + T cell recruitment communications from myeloid populations in ribociclib-resistant versus sensitive tumors (top panels) post-treatment in the discovery/validation cohorts but not under

letrozole alone (HRE statistics in Supplementary Data 29)(color = integrin recruitment communication)(Sample size: as in B). **E** Box plot: reduced T cell abundance in ribociclib-resistant tumors (top) during treatment (x-axis) causing lower abundance post-treatment versus sensitive tumors in discovery/validation cohorts (shape) (logistic regression:logit(resistant-trend):estimate = −0.73,se = 0.046, df = 100,z = −16.06, p = 2e-16;post-treatment:estimate = 0.60,se = 0.058, df = 100,z = 10.47, p = 2e-16). Throughout letrozole alone treatment, T cell abundance was lower in resistant versus sensitive tumors (logit(difference):estimate = 0.61,se = 0.050, df = 59,z = 12.3, p = 2e-16). Sample size: 424,581 cells (T-cells = 3166),173 tumor samples,62 patients,3 timepoints. **F** Box plot: reduced immune response phenotype across cell types (x-axis) post-treatment in resistant versus sensitive tumors under combination ribociclib (top) but not letrozole alone (bottom)(HRE::ribociclib: [Cancer:estimate = −0.027,se = 0.001, df = 1913, t = −20.2, p = 2e-16;Diploid epithelial:estimate = −0.018,se=0.001, df = 2154, t = −12.6, p = 2e-16;Myeloid:estimate = −0.013,se = 0.002, df = 3551, t = −8.49, p = 2e-16;CD8 + T-cell:estimate = −0.026,se = 0.005, df = 6918, t = −5.24, p = 1.7e-7;CD4 + T-cell:estimate = −0.012,se = 0.002, df = 4916, t = −6.03, p = 1.7e-9;Stromal:estimate = −0.014,se = 0.0014, df = 1961, t = −9.9, p = 2e-16];letrozole[Cancer:estimate=0.014,se=0.009, df=1.8, t = 1.47, p = 0.16;Diploid epithelial:estimate = 0.002,se = 0.009, df = 1.9, t = 0.25, p = 0.81;Myeloid:estimate = −0.005,se = 0.009, df = 1.9, t = −0.61, p = 0.55;CD8 + T-cell:estimate = −0.004,se = 0.011, df = 4.1, t = −0.36, p = 0.72;CD4 + T-cell:estimate = −0.014,se = 0.009, df = 1.9, t = −1.58, p = 0.13;Stromal:estimate = −0.005,se = 0.0092, df = 1.8, t = −0.55, p = 0.59]). Points = mean(ssGSEA hallmark interferon gamma response score) per tumor in discovery/validation cohorts (shape). Sample size:108844 cells (Discovery = 43814;Validation = 65030), 53 post-treatment tumors biopsies. All box elements represent median (center line),upper/lower quantiles (hinges),1.5*inter-quartile range (whiskers). All statistical tests two-sided. Source data provided as a Source Data file.

pathway contrasting gene expression of naive and cancer killing effector cells (GSE22886: Naive CD8 T cell vs NK cell up pathway). This analysis showed that activation to an effector CD8 + T cell phenotype was associated with the strength of inflammatory cytokine communications received from macrophages (est = 0.10, df = 1975, t = 7.69,

p < 0.0001) (Figure S15D). In growing ribociclib-resistant tumors, the loss of T cell activating communications was concurrent with the reduction of CD8 + T cell differentiation away from a cytotoxic effector state during treatment (est = −1.33, df=112.7, t = −4.26, p < 0.0001) (Fig. 5C).

By the end of treatment, T cells of growing ribociclib-resistant tumors received fewer recruitment signals from macrophages compared to shrinking ribociclib-sensitive tumors (Supplementary Data 25). Stimulation of a variety of T cell integrin receptors was reduced, including *ITGB1*, *ITGB2* and *ITGA4* (Fig. 5D top left) (*ADAM121-ITGB1*: est = 0.044, df = 293.7, t = 8.33, $p < 0.0001$; *ICAM2-ITGB2*: est = 0.058, df = 281.7, t = 6.57, $p < 0.0001$; *VCAM1-ITGA4*: est = 0.064, df = 461, t = 13.1, $p < 0.0001$). These integrins are critical for T cell migration and recruitment through basement membranes[70,71]. Again, these results were validated in the independently profiled validation cohort, with T cells of growing ribociclib-resistant tumors receiving fewer recruitment signals than those of shrinking ribociclib-sensitive tumors (Fig. 5D top right) (Supplementary Data 26-29). In contrast, myeloid recruitment signaling to T cells was not linked to letrozole response in either the discovery or validation cohorts (Fig. 5D bottom panels). We next confirmed that in growing ribociclib-resistant tumors, the decrease of T cell recruitment communications from myeloid cells was linked to a post treatment reduction in T cell abundance (est = −0.73, df = 100, $z = -16.06$, $p < 0.0001$) (Fig. 5E). In contrast, a stable abundance of T cells was observed in ribociclib-sensitive shrinking tumors. Throughout letrozole treatment, T cell abundance was also lower in growing than in shrinking tumors (est = −0.61, df = 59, $z = 12.3$, $p < 0.0001$). Together, these results indicate the central role of macrophages in orchestrating T cell activation and recruitment and an effective anti-tumor response during ribociclib treatment.

We then assessed how cancer and non-cancer cells responded to the diverse cytokine communications in the TME. We hypothesized that in immune hot TME's, the high levels of immune activating signals and T cell recruitment and activation should induce an inflammatory phenotypic response across cell types, with activation of interferon regulatory factors (IRFs) and induction of interferon-stimulated genes (e.g., interferon gamma-induced proteins) allowing recognition and killing of cancer cells[72,73]. The activation of the interferon gamma response pathway is expected in response to cancer antigens rather than interferon alpha response upon viral infection. We measured the interferon gamma response at the end of treatment in cells of each cell type and across tumors using the Hallmark interferon gamma response ssGSEA signature. The interferon gamma immune response was suppressed across all cell types in growing ribociclib-resistant tumors compared with either ribociclib-sensitive shrinking tumors (Fig. 5F), but this was not observed under letrozole alone.

Cancer cells in particular exhibited a substantially weaker interferon gamma response post treatment in growing ribociclib-resistant tumors compared to ribociclib-sensitive shrinking tumors (Fig. 5F top) (est = −0.027, df = 1913, t = −20.2, $p < 0.0001$). This lack of immune detection in growing tumors was confirmed in ribociclib-resistant cancer cells of the validation cohort (est = −0.006, df = 1459, t = −4.3, $p < 0.0001$) but was not observed in growing letrozole-resistant tumors of either the discover cohort (Fig. 5E bottom) or the validation cohort (est = −0.003, df = 9.6, t = −0.2, $p = 0.84$). Communication pathways strongly associated with high cancer interferon gamma response phenotype activation were found using Lasso regression (see Methods). A cancer interferon gamma response was frequently associated with receipt of strong IL-15 signals (28% of tumor subclones) or related cytokine receptors such as: Toll-like receptor 2 (*TLR2*: 19% subclones), Interleukin-22 Receptor Subunit Alpha 1 (*IL22RA1*: 17% subclones), Interleukin-12 Receptor Subunit Beta-1 (*IL12RB1*: 15% subclones) (Figure S16). Together these results indicate that cells in growing ribociclib-resistant tumors experienced a less hot tumor microenvironment at end of treatment compared to those of ribociclib-sensitive shrinking tumors. This was linked to cancer cells exhibiting a weaker interferon phenotype response, lessening immune detection.

## IL-15 treatment overcomes immune suppressive effects of CDK4/6 inhibition

We next assessed the immune suppressive effect of ribociclib treatment both in patient peripheral blood samples from the FELINE trial and using in vitro experimental model systems coculturing cancer cells with patient-derived T cells.

We first analyzed the white blood cell (WBC) counts in FELINE patient peripheral blood mononuclear cell samples obtained throughout treatment with either combination ribociclib or letrozole alone. We found, using a generalized additive model, that early in treatment the abundance of WBC's was approximately halved under ribociclib treatment (eff.df = 2.27, F = 12.52, p = 4.6e-7) but remained stable under letrozole treatment (Fig. 6A). This independent peripheral blood mononuclear cell data aligns with the immune suppressive side effects of ribociclib identified through our scRNAseq analyses of composition and communication.

We then performed in vitro experiments to validate the direct inhibitory effects of ribociclib on patient-derived CD8 + T cells and examined how CD8 + T cell viability is impacted by IL-15 cytokine signals in the environment in the presence or absence of ribociclib (Fig. 6B). We confirmed that the viability and spheroid area of patient-derived CD8 + T cell populations was reduced by ribociclib treatment (viability: est = −0.48, df = 2, t = −6.04, $p < 0.005$; area: est = −0.44, df = 2, t = −14.8, $p < 0.0001$) but was substantially increased by IL-15 cytokine treatment (viability: est = 0.29, df = 45, t = 8.63, $p < 0.0001$; area: est = 0.23, df = 43, t = 12.4, $p < 0.0001$). Similarly, T cell ATP, and proliferation was reduced by ribociclib treatment but more greatly increased by IL-15 cytokine treatment (Figure S17A/B). CD8 + T cell activation was confirmed by Interferon gamma (IFN-γ) production in monoculture p (<0.05) and coculture with cancer cells when treated with IL-15 ($p < 0.0001$, Figure S17C). In the presence of IL-15, IFN-γ production significantly increased in coculture with cancer cells compared to monoculture alone ($p < 0.0001$) and was unaffected by ribociclib treatment in both monoculture and coculture conditions. This indicates that IL-15 cytokine treatment can rescue CD8 + T cell proliferation and activation during ribociclib treatment and overcome immune suppressive side effects.

We then examined how ribociclib and IL-15 cytokine signals impact CD8 + T cell regulation of cancer population growth by experimentally coculturing patient-derived T cells with one of four fluorescently labeled cancer cell lines (ribociclib resistant vs sensitive CAMA-1 and MDA-MB-134 cells; non-autologous). Resistant cancer cell lines were generated from the parental sensitive line through long term selection. Using serial imaging, we tracked cancer population growth over time in mono and coculture conditions across a six-point gradient of IL-15 concentration and with or without ribociclib treatment (Figure S18). The average growth of each replicate cancer population during 7 days of treatment was measured by the relative growth rate (rgr) (see methods). We confirmed that, despite cancer cell lines being allogeneic, T cells did not show Graft-versus-host disease (GVHD) reactions and had little effect on cancer growth without IL-15 treatment. For each cancer cell line, we then compared the impact of IL-15 treatment on cancer growth in cancer monocultures and cancer-T cell co-cultures between ribociclib treatment and DMSO control conditions (Fig. 6C).

This analysis showed that cancer monocultures were unaffected by IL-15 concentration. In cocultures, IL-15 T cell activation significantly reduced cancer population growth of ribociclib sensitive and resistant cells. Ribociclib had little effect at slowing the growth of resistant cells (Figure S19). Further, the immune suppressive effect of ribociclib on T cells opposed IL-15 T cell activation, reducing the efficacy of T cells in regulating cancer growth (Fig. 6C, D). However, with higher concentrations of IL-15 the T cells effectively controlled

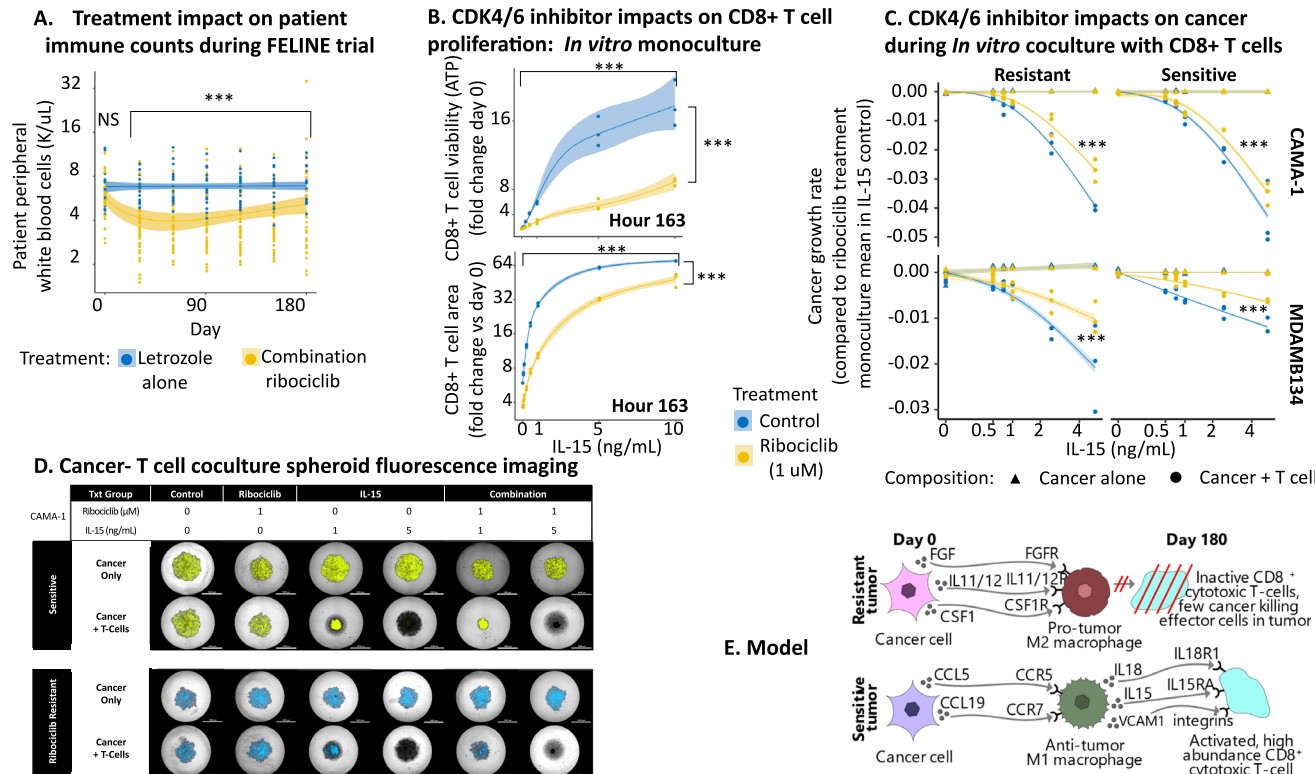

**Fig. 6 | Interleukin 15 (IL-15) addition overcomes CDK4/6i-induced immune suppression, boosting CD8 + T cell activation and cancer control. A** Ribociclib immune-suppression shown by decreased peripheral white blood cell (WBC) counts in serial blood samples (points) of FELINE patients during combination ribociclib treatment but not letrozole alone (generalized additive model (GAM):non-linear trend::ribocicli:eff.df = 2.27, F = 12.52, p = 4.6e-7;letrozole:eff.df = 1.0(linear), F = 0.045, p = 0.83(trend; not significant:NS))(solid line = GAM treatment-specific trend,shaded = 95% confidence interval(+/-1.96*SE)). No pre-treatment difference between arms (log-linear model+ANOVA:estimate = 0.073,se = 0.081, df = 59, t = 0.91, p = 0.37). Sample size = 408 blood draws, 62 patients (Treatment:ribociclib = 39,letrozole = 23) across 7 timepoints. **B** Ribociclib (color) reduced patient-derived CD8 + T cell viability (top) and area (bottom) after 163-hour monoculture (log-linear model(IL-15 = 0 ng/mL):ATP:estimate = −0.48,se = 0.079, df = 2, t = −6.04, p = 0.0038;Area:estimate = −0.44,se = 0.03, df = 2, t = −14.8, p = 0.0045). IL-15 over-came inhibition, with IL-15 > 1 ng/mL restoring viability/area above DMSO control (GAM:IL-15 activation::ATP:eff.df = 3, F = 760.87, p = 2e-16;Area:eff.df = 2.94, F = 611.5, p = 2e-16). GAM characterized non-linear dose-dependent IL-15 effect with/without ribociclib (solid line = expectation;shaded regions = 95% confidence interval (expectation + /-1.96*se)). Sample size = 48 measurements, 3 experimental replicates under 16 treatment (8 IL-15 levels(0-10 ng/mL)+/-1uM ribociclib). **C** IL-15 (x-axis) slowed cancer growth of 4 cell lines (panels:CAMA-1/MDA-MB-134 ribociclib-resistant/sensitive pairs) in patient-derived T cell cocultures (circles;seeding-ratio::-cancer:T-cell=4:1) but not monocultures (triangles)(GAM::IL-15|coculture:CAMA-1:resistant:edf = 2.00, F = 1323.00, p = 1e-16,CAMA-1:sensitive:edf = 2.00, F = 543.25, p = 1e-16,MD-AMB-134:resistant:edf = 2.00, F = 146.85, p = 1e-16,MDAMB134:

sensitive:edf = 1.70, F = 693.80, p = 1e-16). Cocultured cancer growth was reduced less by IL-15 stimulation under ribociclib treatment (yellow circles) versus DMSO control cocultures (blue circles)(GAM::IL-15|ribo:CAMA-1:resistant:edf = 1.88, F = 56.15, p < 1e-16,CAMA-1:sensitive:edf = 1.00, F = 9.61, p = 2.7e-5,MD-AMB-134:resistant:edf = 1.88, F = 16.07, p = 1.5e-6,MDAMB134:sensitive:edf = 1.95, F = 51.77, p < e-16). Higher dose IL-15 controlled ribociclib-treated coculture cancer growth. Points = replicate cancer growth rate over 6 days versus mean of IL-15 untreated monocultures (blue triangles:IL-15 = 0 ng/mL) per ribociclib treatment. GAMs characterized treatment impacts on growth rates per lineage and coculture composition (dashed/solid lines = mono/coculture expectations;shaded = 95% confidence intervals (+/-1.96*se)). Sample size = 288 spheroids, 4 cancer lineages (CAMA-1/MDA-MB-134 resistant/sensitive), 2 compositions (mono/coculture), 6 IL-15 concentrations (0/0.5/0.75/1/2.5/5 ng/mL), 2 ribociclib doses (0/1uM), 3 experimental replicates. **D** Representative florescent imaging (4x magnification) demonstrating IL-15 activation of an effective cytotoxic T-Cell (unlabeled: black) response in sensitive (YFP labeled:yellow) or ribociclib-resistant (CFP labeled:blue) CAMA-1 cancer cells. Cancer monoculture (top rows) and cancer-T cell coculture (bottom rows) spheroids following 6-day treatment with DMSO (0.1%;control), ribociclib (1 μM), IL-15 (1-5 ng/mL), or combination ribociclib (1 μM) + IL-15 (1-5 ng/mL). **E** Schematic: diverse pre-treatment cancer communications stimulate immune-suppressing M2-like myeloid polarization in ribociclib-resistant tumors. Reduced pro-immune M1-like myeloid differentiation diminishes interleukin/integrin signaling and subsequent CD8 + T cell activation/recruitment, preventing effective killing of quiescent cancer cells. All statistical tests two-sided. Source data provided as a Source Data file.

cancer growth, leading to cancer spheroid shrinkage and the essential eradication of both resistant and sensitive cancer cell populations (Fig. 6C, D) (MDA-MB-134 spheroid images in Figure S20). We confirmed, using cancer-macrophages-T cell tricultures, that in presence of monocyte derived macrophages the combination of ribociclib and IL-15 promotes cancer control via T cell killing (Figure S21). These results show that IL-15 activation of T cells can overcome the immune suppressive effects of ribociclib treatment and reinvigorate the control of cancer growth, which is in line with the computational patient-focused tumor analyses.

## Discussion

Analyzes of the phenotypes, composition and communication of cell types within the TME revealed the central role of tumor ecosystem-wide signaling in CDK4/6 inhibitor treatment response (Fig. 6E). Cancer cells from growing ribociclib-resistant tumors modify the behavior of immune regulating myeloid cells, through production of diverse communications including *CSF1*, *CLCF1*, *FGF*, and *TGF* family members and interleukins 5/11/12. These ribociclib-resistant tumor specific communications polarize myeloid cells into an M2-like phenotype, suppressing subsequent T cell activation and recruitment through loss

of interleukin and integrin signaling respectively. These signaling effects precede a reduced T cell differentiation and abundances in growing ribociclib-resistant tumors during treatment and associate with a lack of an interferon response required for cancer recognition and killing by T cells. Assessment of patient's peripheral immune cell counts indicated the systematic immune-suppressive side-effects of CDK4/6 inhibitor treatment. In vitro cancer-immune coculture assays showed that IL-15 treatment can reinvigorate T cell proliferation and cancer control.

Results highlight how the tumor microenvironment impacts CDK4/6 inhibitor response. Immune activation at baseline and during treatment correlated with improved tumor response to both letrozole alone and combination ribociclib treatments. However, response to ribociclib was far more dependent on the composition, phenotypes and communication of immune cells, particularly macrophages, compared to letrozole response. This finding may be due to the broader effects of ribociclib on immune cells, as it is known that cell cycle inhibitors can cause low white blood cell counts and block T cell proliferation[15,17,74]. The dual impact on both cancer and immune cell proliferation could more strongly exacerbate pre-existing TME differences that determine a tumor's response. In contrast, the targeted effect of letrozole on the cancer cell specific estrogen growth factor pathway has less impact on immune cell signaling and T cell cytotoxicity.

Results in Figs. 5, 6 support the hypothesis that recruitment, abundance, and activation of cytotoxic T cells are critical components of CDK4/6 inhibitor efficacy[13]. It may often be insufficient to block cancer cell growth alone; immune cell recognition, cytotoxic effects, and clearance of cancer cells may be a critical component of tumor response to cell cycle therapies. We have previously shown that tumor phenotypic evolution during combination ribociclib therapy leads to the emergence of cell cycle reactivation through a shift from estrogen to alternative growth signal-mediated proliferation in this early-stage ER+ breast cancer population[10]. We propose that a durable tumor response to cell cycle treatment requires the killing of cancer cells by the immune system. Otherwise, it is possible that the cancer cells can evolve to bypass the effects of anti-proliferative drugs.

The perspective that breast cancer is immunologically cold and not treatable with immunotherapy is being challenged[75], yet clinically ER+ tumors are still presumed to be less responsive to immunotherapy than other subtypes[76,77]. However, evidence is accumulating that a strong immune response is essential to ensure tumors respond well to CDK4/6 therapy[14–17,20,21]. For example, a high abundance of inactive or regulatory T cells predicted worse overall survival and relapse risk of metastatic ER+ breast cancer patients during the RIBECCA trial[17] and more generally across other treatments[78]. Conversely, the co-occurrence of many CD8+ cytotoxic T cells with CD4+ helper T cells has been associated with increased progression free and overall survival[79]. Our results in Fig. 2 show that within ER+ breast cancers, there exist three distinct archetypical ecosystem compositions: i) a cancer-dominated state, ii) an immune-hot and diverse state, and iii) a fibroblast and endothelial-enriched state. Those initially immune-hot tumors are sensitive to treatment with anti-proliferative endocrine and cell cycle inhibitor treatments. This result indicates that preemptively or concurrently heating cold tumors may improve response to cell cycle and endocrine inhibition. CDK4/6 inhibitors themselves have some favorable immunomodulatory effects, such as increasing immunogenicity and T cell activation[14–16]. However, results in Fig. 6 highlight their counteracting immunosuppressive side-effects on T cell proliferation, recruitment and activation that predominate in resistant tumors. These dual effects explain the association of CDK4/6 inhibitor clinical responses with a reduction of immunosuppressive cell types in the peripheral blood of metastatic patients[21]. Together these findings indicate that CDK4/6 treatment efficacy can be improved by overcoming immune

suppressive side effects whilst nurturing favorable immune modulation.

A broad range of immune signaling exists that impacts cancer recognition across hot and cold tumors[80]. Our results in Fig. 4 show that cancer to macrophage communications prior to treatment are associated with suppression of immune function in ribociclib-resistant tumors. These data identify immunotherapy targets to overcome cell cycle inhibitor resistance, such as myeloid cell differentiation and cytokine signaling[81]. Our results show that cell cycle therapy response is higher in tumors with more abundant dendritic cells and more inflammatory M1-like macrophages. Several promising immunotherapeutic strategies are emerging to promote M1-like differentiation[82]. These include: local low-dose irradiation[83], intra-tumoral IL-21 injections[84], immune checkpoint inhibitors[85], autologous GM-CSF vaccines[86] or chimeric antigen receptor macrophage transfer[87]. Treatments could directly activate cytokine signaling to recruit and activate effector T cells and promote antigen presentation. Our analyses identified that shrinking ribociclib-sensitive tumors had increased IL-15 signaling between macrophages, T cells and cancer cells, leading to increased T cell activation and stronger antigenic interferon responses. IL-15 T cell activation can overcome the immune suppressive side-effects of ribociclib. In both breast and colon cancer murine models, IL-15 promotes tumor destruction and reduces metastasis through T cell activation[88,89]. Furthermore, IL-15 agonizts can overcome the immunosuppressive effects of anti-proliferative drugs (e.g., MEK inhibitors)[90]. Ongoing clinical studies are testing IL-15 efficacy as an adjuvant or combination treatment for metastatic solid tumors[91]. Potential benefits over other FDA approved cytokines such as IL-2 may include reduced toxicity and avoidance of Treg differentiation or activation-induced CD8 + T cell death[92].

By serially profiling cancer and non-cancer cells in a cohort of patient tumors resistant or sensitive to treatment, we can identify the key mechanisms of resistance driven by the tumor microenvironment (TME). This analysis provides valuable insights into tumor composition, intercellular communication, and phenotypes, which can be used to identify new treatment strategies. By examining the TME composition and myeloid cell phenotypes, we can detect early indicators of tumor sensitivity to cell cycle and endocrine therapies. Moreover, studying the dynamics of communication between cancer and immune cells revealed a key immunological component to ribociclib resistance. It also uncovers potential mechanisms underlying TME dysregulation and identifies possible treatment targets to counteract the immune-suppressive effects of ribociclib. Approaches to reignite immune activity in early-stage ER+ breast cancer may help overcome CDK4/6 inhibitor resistance, enhancing the effectiveness of treatment strategies.

## Methods
### Patient cohort and sample collection
Patient tumor core biopsies were collected prospectively under Clinical Trial #NCT02712723[31], during a randomized, placebo controlled, multicenter investigator-initiated trial led by Dr. Qamar Khan at the University of Kansas Medical Center (IND #127673). The trial entitled FELINE studied Femara (letrozole) plus ribociclib (LEE011) or placebo as neo-adjuvant endocrine therapy for women with ER-positive, HER2-negative early-stage breast cancer. Postmenopausal women with pathologically confirmed non-metastatic, operable, invasive breast cancer and clinical tumor size of at least 2 cm were enrolled from 10 centers across the United States. Invasive breast cancer had to be ER positive (≥ 66% of the cells positive or ER Allred score 6–8) and HER2 negative by ASCO-CAP guidelines.

One hundred and twenty patients were randomized equally across three treatment arms (40:40:40). Arm A received letrozole plus placebo, Arm B letrozole plus ribociclib 600 mg daily for 21 out of 28 days of each cycle and Arm C received letrozole plus ribociclib

400 mg continuously. Protocol therapy was continued until the day before surgery. Tumor response to treatment was assessed using multiple imaging modalities. Mammogram, MRI and ultrasound of the affected breast were performed at baseline and a mammogram and ultrasound was performed at completion of neoadjuvant therapy. MRI of the breast was performed after completion of 2 cycles of treatment (Day 1 of cycle 3). Serial tissue biopsies using a 14-gauge needle were mandatory, providing three core tumor sample over the course of treatment: baseline (Day 0), Cycle 1 follow up (Day 14), and end of treatment (Day 180). Immediately after collection, biopsy samples were snap frozen embedded in optimal cutting temperature conditions. Informed consent was obtained from all patients following protocols approved by the institutional IRBs and in accordance with the Declaration of Helsinki. The study was approved by University of Kansas Institutional Review Board (protocol #CLEE011XUS10T).

### Single nuclei RNA sequencing and processing

Tumor single cell nuclei were isolated from OCT embedded core tumor biopsies using a modified lysis buffer containing 0.2% Igepal CA-630 as previously described[93]. Single cell RNA-Sequencing (scRNAseq) was performed on single nuclei suspensions (i.e., single nuclei RNA sequencing) using 10X Genomics Chromium platform as previously described[10]. Sequence reads were processed with BETSY and CellRanger v3.0.2, which aligned reads to reference genome (GRChg38) using STAR v2.6.0[94]. For each sample, a gene-barcode count matrix was generated containing counts of unique molecular identifiers (UMIs) for each gene in each cell.

We reanalyzed the validation cohort cells to recover intermediate-quality non-cancer cells that were excluded based on the filters used originally in the discovery cohort analysis. Cells were clustered based on the percentage of mitochondrial genes using k-means clustering ($k = 4$). We filtered out high mitochondrial content clusters (centroids = 55 and 87% mitochondrial genes) and retained low percent mitochondrial genes (centers = 0.2 and 20%). We further filtered out cells classified as epithelial by SingleR analysis[33] and with less than 100 genes expressed.

### Cell type classification and verification

We obtained transcriptional profiles of 424,581 single cells, using stringent quality controls to ensure high-coverage, low mitochondrial content, and doublet removal (detailed in ref. [10]). On average, we recovered 2.75 (out of 3) time point samples per patient. Broad cell types were annotated using singleR[33], cancer cells were identified by their frequent and pronounced copy numbers amplification using InferCNV[34]. Cell type annotations were verified by cell type specific marker gene expression and UMAP/TSNE analyses[10,35]. Granular immune subtype annotations were obtained using our recently published ImmClassifier machine learning method, which has been validated by flow cytometry comparisons[36].

### Machine learning classifier for cell type annotations

Cell subtype annotations for the discovery and validation cohorts were confirmed to be consistently annotated by training a random forest machine learning classifier to identify cell types using the well-curated discovery cohort data. We then applied the classifier to predict cell type annotations in the validation cohort. First, we identified the marker genes associated with each cell type in the discovery cohort cells, using a negative binomial test to find genes differentially expressed in each cell type relative to all others. Genes expressed in > 25% of cells in at least one group and showing a log fold change in expression > 0.25 between the groups were selected as candidate markers. We additionally included cell cycle score (G2M and S scores calculated by Seurat's CellCycleScoring function) as latent variables. The top 100 marker genes of each cell type were selected as candidate features in the machine learning analysis. The

classifier was constructed using SingleCellNet, a top pair random forest approach to predict each cell type[95]. First, we split the high-quality discovery cohort into a training and validation subset. We then trained the classifier on the training subset with the parameters nTopGenes = 25, nRand = 100, nTrees = 1000 and nTopGenePairs = 50. Performance assessment in the held-out validation subset showed good performance with area under the receiver operator curve > 0.9 (Figure S1C). To confirm the consistency of the discovery and validation cohorts, all cells were projected into a common UMAP space, using the first 10 principal components of the scaled expression levels of 100 marker genes associated with each cell type (Figure S1). We verified that the UMAP clusters, indicating a major biological cell type, were assigned consistent cell type annotations across cohorts when using the manual curation and machine learning classification approaches. Most cell types were uniquely assigned to a single cluster and this accuracy was further improved by retaining cells with a cell type prediction probability > 0.75.

### Archetypal tumor compositions

The composition of each tumor sample was summarized by first calculating the proportion of each cell subtype, to correct for sampling variation. The compositional similarity of each tumor sample was then measured using the pairwise logit-Euclidean and supported by Manhattan distances. This quantified the fraction of each tumor composition that would need to be altered to generate the compositional profile of each other sample. Compositionally similar tumors and collections of cell types with correlated abundances were grouped using hierarchical clustering (method = 'ward.D2').

To identify archetypal ER+ breast cancer tumor compositions, we projected all tumor samples into a composition landscape, using UMAP (version 0.2.3.1) to account for the non-linearity and non-normality of compositional data[35]. Highly compositionally similar tumors located close together in this ordination space and distant from tumors with divergent compositions. We then applied a Gaussian Mixture model (GMM) and Bayesian information criterion to probabilistically identify distinctly similar clusters of tumor samples[96]. This identified the appropriate number of archetypal tumor compositions supported by the data and classified each tumor sample into an archetype. Major compositional differences between archetypal compositions were identified using Dirichlet regression (R package DirichletRegv0.7-1) and the rank correlation of UMAP TME composition axes with cell type frequencies.

### Subclonal cancer composition and evolution from scRNA copy number alteration

Cancer subclonal populations of each tumor sample were identified through infercnv analysis (R package infercnv v1.0.2; cutoff = 0, min_cells_per_gene = 100 or 500, cluster_by_groups = T, HMM = T, analysis_mode= "subclusters"). Genomic regions of copy number alteration in each cell were detected relative to a subset of 500 reference immune or stromal cells, using the count matrix. Then, cancer populations with distinct copy number profiles were defined as cancer subclones of a patient tumor using hierarchical clustering (R package fastcluster v1.1.25; method = 'ward.D2')[97]. Clusters with distinct copy number profiles were defined as subclones for each patient. Single-cell grouping was performed based on hierarchical cluster analysis.

### Cell phenotypes from Gene Set enrichment analysis

The gene expression count matrix of each cell type was filtered to keep genes expressed in at least 10 cells, zinbwave normalized with total number of counts, gene length and GC-content as covariates (R package zinbwave v 1.8.0; K = 2, X = "-log (total number of counts)", V = " ~ GC-content + log (gene length)", epsilon = 1000, normalizedValues = TRUE)[98]. Single sample Gene Set Enrichment Analysis (ssGSEA) scores of

50 hallmark signatures (MSigDB, hallmark) and 4725 curated pathway signatures (MSigDB, c2) were calculated for each cell using the normalized count matrix in GSVA (R package GSVA v1.30.0; kcdf = "Gaussian", method = 'ssgsea')[99,100].

### Communication from across diverse cell type populations through: Tumor-wide integration of signaling to each receiver cell

Networks of communication from across diverse cell type populations and received by individual cancer and non-cancer cells of a tumor were uncovered by applying an extended expression product method to ligand-receptor scRNAseq data. This measures population level communication using single-cell gene expression (count per million). We first extended individual level cell-cell interaction (CCI) approaches (reviewed in ref. 22) to measure communications received from entire cell type populations or from across the entire tumor population (tumor-wide communication), accounting for tumor composition and within cell type phenotypic heterogeneity. The tumor-wide communication metric was derived by formulating a differential equation model of tumor ecosystem signaling. This describes the change in concentration of a signaling ligand molecule ($S$) in the TME as following:

$$\frac{dS}{dt} = \sum_i \sigma x_i P_i - \sum_j q_j y_j \gamma P_j S - \mu S.$$

Signals are produced by cells in the TME at a rate proportional to their expression of the signaling ligands. Within cancer and non-cancer cell types, subpopulations vary phenotypically and differ in ligand gene expression, with subpopulation $i$ having ligand expression $x_i$. Ligands produced by a cell are released into TME at rate $\sigma$. The total signal production by each cell subpopulation is proportional to heir abundance in the TME ($P_i$) and the total signal production across the TME is given by the sum of production across all cell subpopulations. Signaling ligands are removed from the TME through decay or diffusion at rate ($\mu$) or when bound to a receptor on a receiving cell (receptor binding rate = $\gamma$) and taken up (ligand internalization rate = $q_j$). Phenotypically different subpopulations within cancer and non-cancer cell types have differing receptor concentrations, with receptor density of cell type $j$ depending on its receptor gene expression ($y_j$). The total ligand uptake by each cell subpopulation is proportional to their abundance in the TME ($P_j$) and the fraction of molecules are taken up and removed from the TME once receptor bound. The total signal uptake is given by the sum of uptake across all cell subpopulations.

The steady state analysis of the TME signal concentration is given by:

$$S^* = \frac{\sum_i \sigma x_i P_i}{\sum_j q_j y_j \gamma P_j + \mu}$$

Assuming ligand release after receptor binding ($q_j$ is small), the strength of signal transmitted from all cell subpopulation in the TME (tumor-wide communication) to a focal receiver cell in subpopulation $j$ is given by:

$$C_j\left(x, y_j\right) = y_j \gamma S^* = \left(\frac{\gamma \sigma}{\mu}\right) y_j \sum_i x_i P_i.$$

Given a sampled tumor composition ($\hat{P}_i$ = cell proportion of subpopulation $i$), the tumor-wide communication transmitted via ligand-receptor pathway $k$ to a receiver cell (of type $j$) can be measured given a vector of ligand gene expression for each cell types present ($x_k$), and the receptor expression of the receiving cell ($y_{jk}$) as: $C_{jk}(x_k, y_{jk}) \propto (y_{jk} \sum_i x_{ik} \hat{P}_i)$ (Fig. 1C).

This generalizes the CCI approaches using the ligand-receptor product and expression correlation method (e.g., CCCExplorer, ICELLNET, NATMI, NicheNet and scTensor; reviewed in ref. 22) to the broader tumor ecosystem perspective. Rather than measuring one-one interactions between individual cells, $C_{jk}(x_k, y_{jk})$ measures the many-one communication strength a focal cell receives from the diversity of cells that are releasing communications into the TME. This is again distinct from the many-many mapping of communication implemented to quantify the probability of cell-cell communication between two cell types (e.g., in CellChat)[24]. The extended expression product method therefore allows an assessment of how phenotypically diverse populations of cells contribute communications to the signaling reservoir in the TME to stimulate the receiver cell, accounting for the abundance and ligand production of each signaling phenotype (Fig. 1C).

We validated that by restricting tumor-wide communications to individual level communications between one sending cell of one cell type and another receiving cell, measurements are consistent with individual level cell-cell interactions obtained using the ligand-receptor correlation/expression product method[3] (as used in CCCExplorer, ICELLNET, NATMI, NicheNet and scTensor; reviewed in ref. 22)[3]. The model also shows how tumor-wide communications generalize the established CCI approach to the broader tumor ecosystem perspective. Crucially, instead of just revealing how an individual cell of one cell type communicates with a cell of another type (individual one-one cell crosstalk), the extended expression product method allows an assessment of how a phenotypically diverse population of cells within each major cell type (e.g., macrophages in distinct states) contribute communications to the receiver, accounting for the abundance and ligand production of each signaling phenotype (population many-one cell crosstalk). This is distinct from methods such as CellChat which use cell counts to weight the probability that an individual of the two cell types interact (i.e., frequency of individual one-one cell crosstalk).

### Measuring tumor-wide communications received from diverse cell phenotypes

We applied the extended expression product method to measure tumor-wide signaling from diverse non-cancer cell sub-populations and heterogeneous cancer lineages to receiving cells. We first resolved diverse subpopulations of each cancer and non-cancer cell type (e.g., macrophages in different differentiation states). For each broad cell type we generated a cell-type specific UMAP based on ssGSEA profiles, with the intrinsic UMAP dimensionality determined using the packing number estimator[101]. We then break down each cell type into subtypes of at least 30 cells with coherent phenotypes and of equal interval width along each phenotype axis. This allowed cell types with relatively continuous phenotypic variation, such as macrophages, to be subdivided into an ordered set of cell states along multiple axes of phenotypic heterogeneity and maintains phenotype covariance structure. We then calculated the relative abundance of each subpopulation of each major cell type within a tumor sample ($\hat{P}_i$).

We next used a curated LR communication database[102] to define a set of 1444 LR communication pathways ($C_{jk}(x_k, y_{jk})$) based on known protein-protein interactions. We extracted single-cell expression of a ligand and used mean CPM of a cell subpopulation as a metric of signal production ($x_k$) and the mean CPM receptor expression to quantify signal receipt by a focal cell ($y_{jk}$). We calculated activity of each LR communication pathway ($k = 1{:}1444$) between each pair of sending ($i$) and receiving ($j$) subpopulations ($i \rightarrow j$) within a tumor:

$$C_{i \rightarrow j, k}\left(x_{ik}, y_{jk}\right) = \left(y_{jk} x_{ik} \hat{P}_i\right).$$

## Strength of communication between cell types: contribution and receipt of signals

To obtain communications via LR pathway $k$ between broad cell types, we totaled signals from sending cell type populations (1: $n$ ligand producing subpopulations of a cell type) and averaged signals to receiving cells (across 1:$m$ signal receiving subpopulations of a cell type). We use a weighted average so that the signal to each receiving cell type population is weighted by abundance (Fig. 1C).

$$C_{i_{1:n} \to \bar{j},k}\left(x_{i_{1:n}k}, y_{j_{1:m}k}\right) = \frac{\sum_{z=1}^{m}\left(C_{i_{1:n} \to j_z k}\left(x_{i_{1:n}}, y_{j_z k}\right)\hat{P}_{j_z}\right)}{\sum_{z=1}^{m}\left(\hat{P}_{j_z}\right)}.$$

We refer to this as the strength of communication from a cell type population to a typical cell of another type. For example, the contribution of $n$ heterogeneous cancer cell populations ($Cancer_{1:n}$) to the communication with a myeloid cell of phenotype $z$ via ligand $x$ and receptor $y$ is given by:

$$C_{Cancer_{1:n} \to Myeloid_z, k}\left(x_{Cancer_{1:n}k}, y_{Myeloid_z k}\right)$$
$$= y_{Myeloid_z k} \sum_{i=1}^{n} x_{Cancer_i k}\hat{P}_{Cancer_i}$$

The strength of communication received by a typical myeloid cell in a sample ($\overline{Myeloid}$) from the diversity of cancer cells is given by:

$$C_{Cancer_{1:n} \to \overline{Myeloid}, k}\left(x_{Cancer_{1:n}k}, y_{Myeloid_{1:m}k}\right)$$
$$= \frac{\sum_{z=1}^{m}\left(C_{Cancer_{1:n} \to Myeloid_z, k}\left(x_{Cancer_{1:n}k}, y_{Myeloid_z k}\right)\hat{P}_{Myeloid_z}\right)}{\sum_{z=1}^{m}\left(\hat{P}_{Myeloid_z}\right)}$$

This was repeated for each LR communication pathway between cell types and across tumor samples. Each communication pathway has a distinct potency to modulate cellular phenotype and behavior and so communication pathway scores were standardized (mean = 0, sd = 1) across patients, preventing highly expressed ligand-receptor pairs dominating communications. The average strength of communication from one cell type to another across LR pathways ($\bar{C}_{i_{1:n} \to \bar{j}}$) was measured by the median standardized communication from one cell to another.

## Validation of communication measurements

We validated the method in peripheral blood immune cells, in which communications between cell types are well known and distinctly different from those expected in tumor biopsy samples[103]. Our approach successfully recovered the expected communication network, with myeloid cells having a central role in communicating via cytokine pathways with many cell types (Figure S22). We also validated that we could recover canonical cell type specific communications including receipt of: macrophage colony stimulating factor primarily in myeloid cells, vascular endothelial growth factor (VEGF) in endothelial cell, fibroblast growth factor (FGF) in fibroblasts, epidermal growth factor (EGF) in epithelial cells and C-C chemokine receptor type 5 (CCR5) in T cells (Figure S5). Finally, we compared the measured differences in communication between the resistant and sensitive tumors of the discovery and validation cohort. This verified the high degree of consistency in the signals each cell type received via each LR communication pathway tumor response groups within across the two cohorts ($R^2 = 0.81$) (Figure S23).

## Cell type communication differences between resistant (growing) and sensitive (shrinking) tumors: Bootstrapping randomization comparison

Contrasting communication across many biopsies, rather than within individual samples, provided comparative insights into the evolution of communication during treatment and the cell type communications that distinguish resistant and sensitive tumors. We contrasted the networks of communication between cell types in resistant and sensitive tumors and examined how communications changed throughout treatment with ribociclib or letrozole. We determined the difference in the average strength of communication from one cell type to another ($\bar{C}_{i_{1:n} \to \bar{j}}$) between resistant and sensitive tumors. To identify which cell type communications significantly differed between tumors resistant and sensitive to each treatment, we perform a bootstrapping randomization analysis. We repeatedly shuffled the observed cell type L-R communications across resistant and sensitive tumors to remove any response related structure of the communication network. For 1000 randomized communication networks, the difference in average communication was recalculated. The distribution of communication differences produced by chance in the randomized networks (null model: average communication does not differ between resistant and sensitive tumors) was then compared to the observed difference in the average communication between cell types.

Using the mean and standard deviation of the communication differences in the randomized networks, z statistics were calculated to indicate how much each cell type's communication differed between resistant and sensitive tumors. Randomization $p$-values were calculated by the rank of the observed average communication difference within the distribution of randomized differences between resistant and sensitive communication networks. A Holm's conservative correction for statistical significance was applied to correct for multiple comparisons.

## Divergent communication networks between cell types in resistant (growing) and sensitive (shrinking) tumors

We obtained the expected cell type communication of one cell type to another in resistant and sensitive tumors at each time point of each treatment. This summarized the average strength of communication ($\bar{C}_{i_{1:n} \to \bar{j}}$) across each individual tumor within each response category and treatment time point. Pre-treatment cell type communication networks were then described by directed weighted network graphs, constructed for resistant and sensitive tumors. Cell types were represented by network nodes and the proportional changes in communication were described by the weight of the vertex from one cell type to another (indicated by arrow width). We also calculated the proportional change in expected cell type communication post treatment in resistant and sensitive tumors, relative to the baseline overall average.

## Communication pathway analysis: identifying response related communications

For each LR communication pathway, we contrasted the strength of communication between cell types in tumors growing (resistant) or shrinking (sensitive) during each treatment. We used log-linear regression to describe trends in cell type communication within resistant and sensitive tumors ($\log(1 + C_{i_{1:n} \to \bar{j}, k}(x_{i_{1:n}k}, y_{j_{1:m}k}))$). General communication trends during treatment and changes specific to growing treatment-resistant tumors were detected using likelihood ratio tests. Significant differences in the strength of cell type communication between resistant and sensitive tumors either before or after treatment were identified by using ANOVA on the endpoint data (Day 0 and 180 separately). We accounted for multiple comparisons using false discovery rate (FDR) $p$-value correction. To identify broadly divergent communications between resistant and sensitive tumors before treatment, we enumerated the detected communications sent and received by each cell type.

## Myeloid phenotype landscape reconstruction

The diversity of myeloid phenotypes was examined through UMAP analysis of single cell transcriptional profiles (log(1 + CPM)). Genes with greater than 5% coverage in cells were used. Dendritic cell and macrophage cell subtype annotations obtained from ImmClassifier[36] were overlaid onto the UMAP, confirming consistent identification of distinct cell population between approaches. The M1-like:M2-like phenotype gradient across the UMAP (dimension 2) was identified using the rank correlation of UMAP axes with each gene's expression. Myeloid phenotypic heterogeneity in the verification cohort was characterized using the UMAP model, trained with the discover cohort CPM data, to project myeloid cells of the validation cohort into a consistent myeloid phenotype space. This provided equivalent M1-like:M2-like phenotype scores for each myeloid cell of the validation cohort. Finally, M1-like macrophages were defined as having below average M1-like:M2-like phenotype scores and other macrophages classified into the M2-like phenotype.

## Myeloid lineage differentiation structure

Trajectories of myeloid differentiation were inferred through pseudotime reconstruction, using the slingshot algorithm[104]. First, we identified 21 distinct myeloid cell clusters from the UMAP phenotype landscape, using a Gaussian mixture model and Bayesian Information Criteria. Next, we characterized the global structure of myeloid differentiation lineages (including the trajectory branching number and locations), using slingshot to construct a cluster-based minimum spanning tree (MST) and describe the continuous differentiation trajectory of each lineage with principal curves. We incorporated the known biology, that macrophages differentiate from monocyte progenitors within this analysis by constraining the MST to begin from the monocyte dominated cluster. The positioning of cells along each lineage trajectory was then obtained by orthogonal projection onto the curves to provide lineage specific pseudotime trajectories. Average pseudotime values quantified the relative differentiation of each myeloid cell away from the monocyte state. Myeloid cells were then categorized into a differentiated or undifferentiated state based on their average pseudotime value, using a two-state Gaussian mixture model. The nonlinear division boundary between M1- and M2-like myeloid cells was detected by applying a generalized additive model to the UMAP coordinates of undifferentiated cells. Myeloid polarization was then measured by the linear pseudotime divergence of cells from the M1/M2 division boundary. This polarization metric provided a continuous measure of myeloid differentiation from a monocyte to either an M1- or M2-like state.

## Detecting the transcriptional profiles defining myeloid polarization

To identify the core transcriptional program defining the M1- to M2-like myeloid cell transition, we characterized the underlying gene expression changes as cells are polarized along the M1/M2 pseudotime trajectories from the monocyte progenitor state. For each gene, we fitted a general additive model (GAM) to detect the potentially nonlinear relationship between polarization pseudotime and gene expression across myeloid cells (following[105]). We identified genes that exhibited significant changes in expression along polarization pseudotime and explored the top 40 genes that were most significantly (greatest F value) linked to increased and decreased polarization.

## Demonstrating myeloid differentiation using knowledge-based gene signatures

To demonstrate that myeloid differentiation and polarization scores reflect an M1- to M2- like transition, a knowledge-defined gene signature was used to test for gene set enrichment (GSE) with inferred myeloid differentiation. Current knowledge of cancer-associated myeloid cell markers from recent scRNAseq analyses across human cancer types were obtained[106,107] and enriched with differentiation markers identified in other recently published studies (Supplementary Data 30). Gene set enrichment (GSE) of single myeloid cells was measured using two comparable methods: Gene Set Variation Analysis (GSVA), a GSE method that estimates variation of gene set activity over a sample population in an unsupervised manner and Pathway Level Analysis of Gene Expression (PLAGE), a GSE method that estimates single cell pathway activity using an SVD-based approach[99,100]. We then examined the relationship between these single cell gene signature scores and the data-derived axes of myeloid differentiation and polarization, using generalized additive models and correlation assessment. We additionally tested for the increased expression of established M2-like macrophage marker genes in cells identified from our data-driven phenotype and pseudotime polarization analyses as having M2-like differentiation.

## Verifying that myeloid polarization predicts resistance

To test for differences in macrophage differentiation between growing treatment-resistant and shrinking treatment-sensitive tumors, we fitted a hierarchical linear model describing how the M1-like:M2-like phenotype score differed in myeloid cells from resistant and sensitive tumors, accounting for patient specific heterogeneity in myeloid phenotype and the shared TME of cells within a sample. We similarly assessed whether myeloid cells of growing ribociclib-resistant tumors showed greater M2-like differentiation in the discovery and validation cohort.

We next confirmed that ribociclib-resistant tumors can be identified early in treatment (Day 0-14) by the increased M2-like differentiation of their myeloid cells. We contrasted the phenotypes of myeloid cells in resistant and sensitive tumors in the independently profiled verification cohort. We applied two complementary analyses. Firstly, we summarized the mean M1-like:M2-like phenotype score of myeloid cells in each tumor. We then contrasted the mean myeloid differentiation early in treatment (Day 0-14) between resistant and sensitive tumors using ANOVA. We confirmed that findings did not depend on the choice of myeloid differentiation metric (UMAP dimension, polarization pseudotime, PLAGE or GSVA). Secondly, we compared the relative abundance of M1-like and M2-like macrophages in resistant and sensitive tumors early in treatment, using logistic regression to describe how the proportion of M1-like cells per tumor biopsy varied by treatment and resistance outcome.

## Measuring targeted cancer cell signaling to M1-like and M2-like macrophages

We identified cancer cell communications that predominantly target either M1-like or M2-like macrophages. First, we measured cell-cell interactions between phenotypically diverse cancer and macrophage subpopulations in each tumor sample, using the ligand-receptor product approach[22]. For each tumor sample, we calculated the average cell-cell interaction of cancer cells with each macrophage via each communication pathway. We contrasted the log cancer-macrophage cell-cell interaction received by M1-like and M2-like macrophages, using a hierarchical linear model to detect differential communication with myeloid cell types and to account for baseline tumor specific differences in cancer-macrophage communication. Significant differences in cancer communication with M1-like and M2-like cells were identified using likelihood ratio tests contrasting: i) the full model with communication to M1-like and M2-like cells differing and ii) the nested null model with no difference in communication. The twenty most significantly activated communications with M1-like and M2-like macrophage were assessed.

### Contrasting the heterogeneity of cancer to macrophage communications across resistant and sensitive tumors

We combined the list of communication pathways through which cancer cells: a) communicate more strongly with macrophages in resistant than sensitive tumors (see Communication pathway analysis) and b) have stronger cell-cell interactions with M2-like versus M1-like macrophages (as described above). From this list, we identified the ligands the cancer cells used to modulate macrophage phenotype and tumor response. Communication pathways binding these ligands were defined as M2-like differentiation communications and selected for supervised analysis. We contrasted the strength of communication from cancer to myeloid cells via each M2-like differentiation communication pathway in resistant and sensitive tumors samples taken early in each treatment (Day 0 and 14). Heatmaps were used to visualize the heterogeneity of communication pathway activity across tumors.

### Cancer and non-cancer cell type contributions to myeloid polarizing communications

We next determined which cell types most strongly contributed to myeloid differentiation communications. We extracted the strengths of each M2-like differentiation communication sent from each cell type to myeloid cells. For each tumor sample, the median standardized M2-like differentiation communication from each cell type was calculated. We then contrasted the average M2-like differentiation communication sent by each cell type in resistant and sensitive tumors and under each treatment.

### Identifying differential communications of M1-like and M2-like myeloid cells with CD8 + T cells

We next identified myeloid communications with T cells primarily produced by either M1-like or M2-like myeloid cells. We first measured cell-cell interactions between phenotypically diverse macrophage and T cell subpopulations in each tumor sample, using the ligand-receptor product approach[22]. For each tumor sample, we calculated the average cell-cell interaction of M1-like and M2-like myeloid cells with T cells via each communication pathway.

We then identified communication pathways by which T cells received significantly different cell-cell interactions from M1-like and M2-like myeloid cells. The log macrophage-T cell interactions from M1-like and M2-like macrophages were contrasted, using hierarchical linear models. A patient specific random component accounted for the heterogeneity in immune communication between TME's and a random component associated with T cell phenotype accounted for the diversity of T cell activation phenotypes within and between tumors. Significant differences in T cell communication from M1-like and M2-like cells were identified using likelihood ratio tests contrasting: i) the full model with communication from M1-like and M2-like cells differing and ii) the nested null model with no difference in communication.

### Contrasting M1-like communication with T cells in resistant and sensitive tumors

Next, we isolated the M1-like macrophages and examined their communication with T cells in resistant and sensitive tumors. For each communication pathway, we contrasted M1-like macrophage to T cell interactions (log transformed) in resistant and sensitive tumor using hierarchical linear models. Again, a patient specific random component accounted for the heterogeneity in immune communication between TME's and a random component associated with T cell phenotype accounted for the diversity of T cell activation phenotypes within and between tumors. A likelihood ratio test was used to detect communication pathways significantly differing between resistant and sensitive tumors.

### Diverging inflammatory communication from myeloid cells to CD8 + T cells in resistant and sensitive tumors

We next determined how the M2-like polarization of the myeloid population in ribociclib-resistant tumors impacted the communication of immune cytokine signals to T cells. First immune activating inflammatory cytokine communications were identified, using the receptors gene-ontology database signatures[69]. For each CD8 + T cell we totaled the signal received from all myeloid subpopulations within that tumor sample via each inflammatory cytokine communication pathway. To obtain the overall immune activating communication received by each CD8 + T cell from the myeloid population, we averaged across communications pathway scores after scaling and log transformation.

We then analyzed at the single cell level how each of the immune activating communications from myeloid to CD8 + T cells diverged during treatment in resistant and sensitive tumors. Using a hierarchical regression model, we described pre-treatment differences in CD8 + T cell activating communication between resistant and sensitive tumors and temporal change during treatment (as previously described in ref. 10). Significant divergence in immune activating communication with T cells of resistant and sensitive tumors was determined using a two-tailed t-test. The Satterthwaite method was applied to perform degree of freedom, t-statistic and $p$-value calculations lmerTest R package[108].

### Linking myeloid inflammatory communications to CD8 + T cell activation

We characterized how the differentiation and activation to an effector CD8 + T cell phenotype was related to the strength of immune cytokine communication they received from myeloid cells. For each CD8 + T cell, differentiation was measured using a CD8 + T cell specific ssGSEA pathway contrasting gene expression of naive and cancer killing effector cells (GSE 22886 Naive CD8 T cell vs NK cell up). The single cell activation state was then linked to the immune activating communication received from across the myeloid population (measured above). Each CD8 + T cells differentiation state was then linked to the inflammatory cytokine communication it received from across the myeloid population. Linear regression was used to measure the increase in T cell activation with increasing inflammatory communication. The strength of inflammatory cytokine communication received was discretized into deciles of signal strength and the distribution of phenotypic state assessed in cells receiving each level of stimulus.

We then analyzed at the single cell level how the CD8 + T cell activation diverged during treatment in resistant and sensitive tumors. Using a hierarchical regression model, we described pre-treatment differences in CD8 + T cell activation between resistant and sensitive tumors and temporal change during treatment (detailed in ref. 10). Significant divergence of CD8 + T cell activation in resistant and sensitive tumors was determined using a two-tailed t-test. The Satterthwaite method was applied to perform degree of freedom, t-statistic and $p$-value calculations, using ref. 108.

### Contrasting T cell relative abundance during treatment in resistant and sensitive tumors

Differences in T cell abundance between resistant and sensitive tumors were analyzed at each treatment time point and separately for tumors receiving each treatment, using logistic regression. We identified significant differences in the proportion of T cells between tumor response groups using a two-tailed Wald-test to generate z statistics and $p$ values.

### Comparing post treatment immune response across cell types in resistant and sensitive tumors

We compared the difference in immune response observed across all cell types between tumors resistant and sensitive to each treatment.

For each single cell observed at the end of treatment, we measured immune stimulation using Hallmark Interferon Gamma response ssGSEA pathway scores.

We analyzed the difference in Interferon Gamma response between treatment-resistant and sensitive tumors, using a nested hierarchical regression model to account for the patient specific differences in immune response and the between cell type differences in this phenotype. Cell type specific random effects were nested within the patient random component, reflecting the occurrence of each cell type within different patient tumors. Significant divergence in Interferon Gamma response between resistant and sensitive tumors treated with combination ribociclib or letrozole alone were determined using two-tailed t-tests.

### Linking inflammatory communications to cancer interferon gamma response

We next determined how cancer cell phenotypes responded to increasing inflammatory cytokine communications in the TME. We assessed the cancer cells interferon response phenotype, using their Hallmark Interferon Gamma Response ssGSEA scores. This cancer phenotype measured intracellular transduction of cytokine signals to the nucleus and induction of interferon regulatory factors (IRFs), interferon-stimulated gene activation (e.g., Interferon gamma-induced proteins) and the production of antigen presenting major histocompatibility complex molecules (MHC I) allowing recognition and killing of cancer cells[72,73].

To determine the major communication pathways stimulating a cancer cell interferon gamma response, we next calculated the strength of the communication each cancer cell received from across the TME via each ligand receptor pathway. For each cancer cell, we coupled the single cell interferon response phenotype to the total communication stimulus each cancer received from across the TME via each receptor. The cancer cell data was subset by subclonal cancer genotype (identified in ref. 10) and communication scores were square root transformed, scaled and centered (mean = 0, sd = 1) to improve normality and comparability respectively.

For each cancer subclone of each tumor sample, we identified communications strongly associated with activation of an interferon response, using a lasso penalized likelihood regression model (R package glmnet). The lasso penalty ($\alpha = 1$) encourages detection of the communications most strongly activating interferon response, through a shrinkage of the coefficients of all but dominant communications predictors. This variable selection approach minimizes overfitting when considering the role of many communication pathways and enhances the interpretability and predictive accuracy of the model.

Cross-validation (internal 10-fold) was performed to determine the penalty parameter ($\lambda$) that minimized the mean cross-validated error. The contribution of each communication to (coefficients) the explained variance in cancer cell interferon phenotype was then assessed. We identified the communication receptors of cancer cells detected to contribute to the interferon phenotype in more than 10% of tumor subclones.

The association of a cancer interferon response with TME communication to cancer via the most frequently detected receptor (IL-15RA) was examined using a generalized additive model with a unique smoothing term for resistant and sensitive tumors given each treatment. The IL-15 communication received by cancer cells was also discretized into deciles of signal strength and the distribution of cancer interferon response phenotypes compared to the signal received.

### CD8 + T-Cell isolation and activation

Leukocyte Reduction System (LRS) cones were obtained from a healthy blood donor at City of Hope, Duarte, CA under Institutional Review Board (IRB # 17387) approval. Blood from LRS cones was transferred to $K_2$EDTA blood collection tube (BD Biosciences), and centrifuged for 10 min at room temperature and $800 \times g$. Plasma was removed and buffy coat was collected and diluted to 5 mL in 1x PBS without $MgCl_2$ (Gibco) + 2% hiFBS (heat inactivated Fetal Bovine Serum). Red blood cells were removed by immunomagnetic depletion using 50 μL of EasySep RBC Depletion Reagent (Stemcell Technologies) per mL of sample according to manufacturer instructions and froze as viable peripheral blood mononuclear cells in 50% RPMI-1640 (Gibco) + 40% heat inactivated FBS (hiFBS, Sigma-Aldrich) + 10% DMSO (Fisher Scientific) and stored in liquid nitrogen vapor phase. Subsequently, CD8 + T cells were isolated from buffy coat using EasySep Human CD8 + T Cell Isolation Kit (Stemcell Technologies) by immunomagnetic negative selection according to manufacturer instructions.

Isolated CD8 + T cells were centrifuged for 5 min at room temperature, $300 \times g$. CD8 + T cells were then resuspended to 0.5e6 cell/mL in RPMI-1640 (Gibco) + 10% hiFBS + 1x antibiotic-antimycotic (Gibco) and stimulated for activation for 4 days supplemented with 20 ng/mL IL-2 (Miltenyi Biotec) and CD3/CD28 Dynabeads Human T-Activator (Gibco) in a 6-well tissue culture treated plate (Corning) and maintained in 37 °C humidified incubator + 5% $CO_2$. Prior to co-culture with cancer cells, CD8 + T cells were collected and CD3/CD28 beads removed using DynaMag-15 magnet (Gibco). Purified activated CD8 + T cells were centrifuged at $300 \times g$ 5 min, at room temperature and resuspended to > 2.0e6 cell/mL in fresh RPMI-1640 complete culture media.

### T cell viability assay

To assess T cell proliferation in the absence of cancer cells, activated CD8 + T cells were cultured in RPMI-1640 + 10% hiFBS + 1x antibiotic-antimycotic with control (0.1% DMSO), 0.5-5 ng/mL IL-15 (R&D Systems), 1 μM ribociclib (Selleck Chemicals), or combinations 0.5-5 ng/mL IL-15 + 1 μM ribociclib in ULA spheroid plates. T cell growth was monitored using Cytation 5 by brightfield imaging every 12 h. After 163 h, proliferation was assessed by measuring total ATP using the CellTiterGlo Luminescent Cell Viability Assay (Promega Corporation). We analyzed how T cell viability was impacted by ribociclib (1 μM) and IL-15 cytokine (0.5-10 ng/mL) treatments, individually or in combination. A generalized additive model characterized the effect of each treatment and combination on the total ATP at 163 h (fold change relative to hour 0). Significant treatment effects on T cell viability were determined using two-tailed t-tests.

### Breast cancer cell line culture

Ribociclib sensitive Estrogen-receptor- positive (ER + ), HER2- breast cancer cell lines CAMA-1 and MDA-MB-134, were respectively cultured in DMEM (Gibco) + 10% hiFBS + 1x antibiotic-antimycotic, or RPMI-1640 + 10% hiFBS + 1x antibiotic-antimycotic. The CAMA-1 ribociclib resistant cell line was established by continuous treatment of 1 μM ribociclib for 1 month followed by 250 nM ribociclib for 4 months as previously described[109]. The MDA-MB-134 ribociclib resistant cell line was established by continuous treatment with increasing concentration of ribociclib from 100 nM up to 500 nM over the course of six months (Figure S24). CAMA-1 (CAMA-1_Sens_V2) and MDA-MB-134 (MDA-MB-134_Sens_V2) sensitive cell lines were fluorescently labeled by transduction with Venus containing lentivirus (LeGO-V2, Addgene #27340, RRID:Addgene_27340), while CAMA-1 ribociclib resistant (CAMA-1_RiboR_Cer2) and MDA-MB-134 ribociclib resistant (MDA-MB-134_RiboR_Cer2) were transduced with Cerulean containing virus (LeGO-Cer2, Addgene #27338, RRID:Addgene_27338). CAMA-1 ribociclib resistant cell line stock culture media was supplemented with continuous treatment of 250 nM ribociclib, while MDA-MB-134 ribociclib resistant cell line was supplemented with 500 nM ribociclib. Fluorescent cell labeling allowed cancer abundance to be quantified in monoculture or when co-cultured with T cells, which were unlabeled. Cancer cell lines were confirmed negative for mycoplasma contamination using MycoAlert PLUS Mycoplasma detection kit

(Lonza) and cell-lines authenticated by STR profiling. All cell lines were cultured and maintained in a 37 °C humidified incubator + 5% $CO_2$.

## Cancer– T cell spheroid co-culture

Cancer cells of each of four cell lines (ribociclib resistant/sensitive CAMA-1 and MDA-MB-134 cells) were plated at 5000 cells per well in a total volume of 100 μL in 96-well Black/Clear Round Bottom Ultra-Low Attachment (ULA) Surface Spheroid Microplate (Corning) in respective cell line complete culture media. After 24 h of spheroid formation, spheroids were imaged prior to sequential addition of 50 μL of T-cell suspension and 50 μL of 4X cytokine treatment to achieve 3 replicates per treatment and coculture group. In coculture treatment replicates, we added 50 μL of 2.5e4 CD8 + T cell per mL medium (final of 1,250 CD8 + T cells per well) (cancer + CD8 + T cell co-cultures) to achieve a (4:1) Cancer: T-Cell Ratio. 50 μL of 4X drug stocks to achieve the following final 1X cytokine/ribociclib treatment combinations in respective cell line complete culture media: control (0.1% DMSO), 0.5, 1, 5 ng/mL IL-15 (R&D Systems), 1 μM ribociclib (SelleckChem), or combination 0.5, 1, or 5 ng/mL IL-15 + 1 μM ribociclib. Cancer – T cell co-culture plates were maintained in a 37 °C humidified incubator + 5% $CO_2$. After 3 days of treatment, a 50% media change was performed with fresh 1X drug stocks using Fluent 780 automated workstation (Tecan).

## Interferon-gamma co-culture assay

Sensitive CAMA-1(CAMA-1_Sens_V2) cancer cells were plated at 5,000 cells per well in 96-well ULA plates and allowed 24 h for spheroid formation incubated in 37 °C incubator + 5% $CO_2$. After 24hrs, 5,000 CD8+ isolated T-cells were added at 50 μL per well to achieve a (1:1) Cancer: T-Cell ratio, and 50 μL of 4x drug stocks were added to achieve final concentrations of 0.1% DMSO, 1 μM Ribociclib, 5 ng/mL IL-15, or combination 1 μM ribociclib + 5 ng/mL IL-15. After 3 days of treatment, 150 μL of cell culture supernatant was collected and placed into 96-well V-bottom plate (Thomas Scientific). 96-well plate was centrifuged for 10 min at $1000 \times g$ to pellet remaining cells and debris. 100 μL of supernatant was collected and Interferon-gamma levels were measured according to manufacturer instructions using the Human Interferon-gamma ELISA kit (Abcam) at $OD_{450nm}$ and $OD_{620nm}$.

## Cancer – macrophage – T cell spheroid tri-culture

THP-1 monocyte cell lines were plated at 2 M cells per 100 mm tissue culture dish and differentiated to M0-like state in RPMI-1640 Media + 10% hiFBS + 1x antibiotic-antimycotic supplemented with 100 ng/mL phorbol 12-myristate 13-acetate (PMA, Sigma-Aldrich) incubated for 72hrs. After differentiation, THP-1 M0-like cells were washed 2x with fresh RPMI culture media and rested for 24hrs prior to initial cancer-macrophage coculture with CAMA-1_SensV2 cells. THP-1 M0 like macrophages and CAMA-1 sensitive cells were collected with 0.25% Trypsin EDTA (Gibco). Cancer cells only or cancer – macrophage coculture (2:1) 5,000 CAMA-1_Sens_V2 cells and 2,500 THP-1 M0-like macrophages were plated in 96-well ULA plates in DMEM + 10% hiFBS + 1x antibiotic-antimycotic and cocultured for 7 days with 50% media change after 4 days. 48hrs prior to cancer – macrophage – T cell tri-culture, CD8 + T cells were isolated and activated with IL-2/CD3/28 as described above. After 7 days of cancer – macrophage coculture, 2,500 CD8 + T-cells were added per well to achieve initial plated cell number ratios of (2:1:1) cancer – macrophage – T cell for tri-cultures in addition to monocultures of cancer cells, T-cells, and cancer- macrophage only cultures. At the time of CD8 + T cell addition, spheroid cultures were concurrently treated with control (0.1% DMSO), 0.1, 0.5, or 1.0 ng/mL IL-15 +/− 1 μM ribociclib and cultured for 6 days. All cell lines were cultured and maintained in a 37°C humidified incubator + 5% $CO_2$.

## Spheroid imaging

Cancer cell abundance was observed throughout treatment by imaging monocultures and coculture spheroids using a Cytation 5 cell imaging multimode reader (BioTek Instruments) with 4X objective under dual Bright Field, YFP (ex 500 / em 542 for Venus florescence), and CFP Fret V2 (ex 433 / em 475 for Cerulean) image acquisition using a 2 × 2 montage image with 5-slice, 50 μm z-stack using Gen5 software (BioTek Instruments, version 3.10.06). Spheroids were imaged prior to co-culture or tri-culture addition or ribociclib/IL-15 treatment and then every 24–72 h post ribociclib/IL-15 treatment for up to 7 days of co-culture or tri-culture. Raw images were analyzed using Gen5 software including image stitching, Z-projection using focused projection, and spheroid size analysis assessing cancer cell viability as measured by total YFP and CFP signal intensity above minimum background intensity thresholds within the calculated spheroid area. Minimum background intensity thresholds for each cell line were confirmed by identifying peaks in the pixel intensity distribution across images that correspond to either viable labeled cell fluorescent signal or background exposure noise using a Gaussian Mixture Model (Figure S25). Fluorescence intensities were then converted into cancer cell type abundances over time after constructing an experimental standard curve to map known cell abundances to fluorescence intensities for each cell line, using linear regression.

## Spheroid co-culture growth response modeling

For each replicate cancer population, we quantified its speed of growth or shrinkage over a 75-hour period. Specifically, the relative growth rate (*rgr*) of each cancer population was determined by the average hourly change in log cancer abundance between the start ($t_0$) and end of treatment ($t_{75}$), calculated as: $rgr = \frac{\log(N(t_{75})) - \log(N(t_0))}{t_{75} - t_0}$ (described in ref. [110]).

We analyzed how the cancer growth rate (*rgr*) of each of the four cell lines was impacted by the individual and combined effects of: i) co-culturing with T cells +/− macrophage cells, ii) cell cycle inhibitor treatment with ribociclib and iii) cytokine treatments with IL-15, using generalized additive models (GAMs) with the following predictor and error structure:

$$rgr_i = \beta_0 + (\beta_{1,M}R + \beta_{1,C}RC) + (s_{2,M}(I) + s_{2,C}(IC)) + (s_{3,M}(IR) + s_{3,C}(IRC)) + e_i$$

$$e_i \sim N(0, \sigma^2)$$

For each of the four cell lines, the fitted GAM estimated the impact on cancer growth rate of: 1) ribociclib treatment in mono/cocultures ($\beta_{1,M}/\beta_{1,C}$), 2) IL-15 concentration in mono/cocultures ($s_{2,M}/s_{2,C}$), and 3) ribociclib and IL-15 synergy in mono/cocultures ($s_{3,M}/s_{3,C}$). For the IL-15 components (terms 2 and 3) we used shrinkage thin plate regression splines to jointly describe the smooth nonlinear dose dependent effects of IL-15 and to perform automatic feature selection and model simplification. For each of the four cell lines, the benefit of IL-15 T cell activation was examined by of the significance of the effect of IL-15 concentration on rgr in cocultures ($s_{2,C}(IC)$). The impact of ribociclib treatment on this T cell killing effect was then determined by the significance of the coculture ribociclib and IL-15 synergy term ($s_{3,C}$). Models were fitted and significance determined using the mgcv r package (version 1.9-0). The significance of treatment effects, co-culturing effects and synergies were determined using two-tailed F-tests.

## Reporting summary

Further information on research design is available in the Nature Portfolio Reporting Summary linked to this article.

## Data availability

Pre-processed single cell RNA-seq gene expression data and relevant metadata are available through GEO (the Gene Expression Omnibus) under accession code GSE211434 at https://www.ncbi.nlm.nih.gov/geo/query/acc.cgi?acc=GSE211434. Source data provided with this manuscript include PBMC data and in vitro T cell coculture cancer growth data collected under ribociclib and IL-15 treatment. Source data are provided with this paper.

## Code availability

Custom code used in analyses and to produce Figs. 1–6 are available on GitHub at https://github.com/U54Bioinformatics/FELINE_project/FELINE_immune_communication.

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

## Acknowledgements

We thank the anonymous patients from the trial that made this study possible. We thank Anne O'Dea, Priyanka Sharma, Cynthia Ma, Meghna Trivedi, Kevin Kalinsky, Kari B. Wisinski, Ruth O'Regan, Issam Makhoul, Laura M. Spring, Aditya Bardia, Yuan Yuan, Lauren Nye, Onalisa Winblad, Jamie Wagner-Berbel, Kelsey Larson, Christa Balanoff, Gregory Crane, Fang Fan, Allison Aripoli, Amanda Amin, Richard McKittrick, Marc Hoffmann, Marc Inciardi, Cory Bivona, Mia Hard, Manana Elia, and Mark Redick for conducting the trial and contributed patient samples. We thank Adam Cohen and Brad Nelson for respectively providing clinical and immunological insights. We thank Eleni Farmaki and Vince Grolmusz for providing lentiviral labeled cell lines, Kimya Karimi for supporting in vitro experiments, Benjamin Copeland for assistance in patient biopsy sample processing and Ben Decato for initial analysis of the raw scRNAseq data. J.G., A.H.B., P.A.C. A.N., F.A. and J.C. were supported by the National Cancer Institute of the National Institutes of Health (NIH) under award number U54CA209978 and U01CA264620. The content is solely the authors responsibility and does not necessarily represent the official views of the NIH. The High-Throughput Genomics Shared Resource was supported by the NIH Award Number P30CA042014. J.T.C. was supported by a Cancer Prevention Research Institute of Texas Core Facility Support Award (RP170668). The authors also thank JKTG for providing funding for this research.

## Author contributions

J.I.G. contributed study design and coordination, performed mathematical/statistical analyses to evaluated tumor composition, cellular phenotypic heterogeneity and tumor-wide communications related to patient outcomes using scRNAseq data, analyzed in vitro experimental data and wrote manuscript. P.C. performed scRNAseq and in vitro experiments and wrote manuscript. E.C.M., J.C., and A.N. conducted bioinformatics pipelines to process scRNAseq data and performed normalization and cell type classification. F.R.A. developed analyses and models and contributed to writing the manuscript. J.T.C. developed bioinformatics pipelines, performed data management and curation and wrote the manuscript. Q.J.K., conceived and coordinated the clinical trial, contributed clinical support and infrastructure and provided clinical data and patient samples as well as contributed to writing the manuscript. A.H.B. designed the research project and analyses, performed scRNA-seq experiments and data analysis, coordinated genomic and mathematical/statistical analyses, and wrote the manuscript.

## Competing interests

Qamar Khan declares research funding from Novartis. Andrea Bild is a founder of Unravel Genomics, which builds biomarkers of drug response. Aritro Nath is co-founder of Unravel Genomics. All other authors have no conflicts of interest to disclose.
