## [Transparent Peer Review file · Nature Communications]

Cellular interactions within the immune microenvironment underpins resistance to cell cycle inhibition in breast cancers

Corresponding Author: Dr Jason Griffiths

Version 0:

Reviewer comments:

Reviewer #1

(Remarks to the Author)

In this study, the authors explore the molecular and cellular crosstalks between cancer cells and immune cells in 2 cohorts (discovery and validation) of ER+PR+ breast cancer patients receiving endocrine therapy (letrozole) alone or combined with CDK4/6 cell cycle inhibition (ribociclib). To this end, authors generated an extensive scRNAseq atlas comprising cancer cells, immune cells and stromal cells of treatment naïve patients, 14 days following treatment and at the end of the treatment (180 days). They use interactome inference to describe how the tumor microenvironment is orchestrated and how it relates to tumor progression and response to treatments. Their analysis show that:

- (1) resistance to combination treatment is associated with M2 macrophage polarization
- (2) in treatment resistant tumors, cancer cells increase communication with M2 macrophages
- (3) response to treatment can be predicted by the immune infiltrated archetype of tumors
- (4) in responders, effector T cells communicate with M1 macrophages
- (5) T cell suppression caused by ribociclib is overcome by addition of IL15 in an organoid approach co-culturing cancer cells and T cells

The study is interesting and provide an important resource for breast cancer research. However, major concerns need to be addressed to strengthen the current claims

In Figure 4:

Fig4.A: Authors acknowledge the diversity of myeloid cells in the TME but keep a very shallow analysis by only clustering apart monocytes, macrophages and dendritic cells. Authors should push their clustering to highlight how macrophage heterogeneity varies between treatments and archetypes.

It is unclear how authors define M1 and M2 macrophages. Authors should use well-defined gene signatures scores for this macrophage states and show how these signatures are enriched by doing GSEA. Such signature can be found in the following studies PMID: 29961579 and PMID: 33545035. Authors should look and show canonical M1/M2 markers including Arg1, MRC1, CD80, TNF etc...

In the same lines, authors should provide the transcriptional program defining M1 anti-tumor macrophages.

Overall, authors should use state of the art knowledge of the cancer-associated myeloid cell field so readers can integrate these data with the current knowledge. In particular the M1/M2 polarization dogma has been extensively revisited in light of the scRNAseq data performed in various human cancers. Several studies have shown that the functional dichotomy does not necessarily align with the phenotype of M1 and M2 macrophages.

Authors claim that treatment resistant tumor promote M2 differentiation while sensitive tumor promote M1 differentiation. To prove that point, authors should perform a trajectory analysis at time 0, 14 and 180 to confirm the differentiation of myeloid cells towards a M2-like or M1-like profile in resistant or sensitive tumours respectively.

In Figure 6:

It is unclear why the co-cultures do not include myeloid cells since authors claims these are the main player in the immunosuppression of T cells. In addition, this co-culture experiments could help address the claim made by the authors that treatment resistant tumors drive the differentiation of monocytes toward the M2 phenotype.

In those lines, authors should:

1- Co-culture monocytes isolated from PBMC of healthy donors with ribociclib-sensitive or -resistant spheroids for 7 days and assess the impact of co-culture on the differentiation of monocytes into macrophages and the polarisation of macrophages. (To confirm in vitro that ribociclib-resistant tumour cells induce the differentiation and polarisation of monocytes into immunosuppressive M2 macrophages)

2- Co-culture sensitive or resistant tumour cell spheroids with monocyte derived macrophages and activated T cells +/- Ribociclib +/- IL-15. (To confirm in vitro that ribociclib IL-15 combination can promote T cells spheroid killing in presence of monocyte derived macrophages polarized tumor cells.

T cell activation, survival and exhaustion should be evaluated within the different conditions.

(Remarks on code availability)

Reviewer #2

(Remarks to the Author)

The paper titled "Breast cancer cells communicate with macrophages to prevent T cell activation during development of cell cycle therapy resistance" is a comprehensive and well analyzed effort to understand the interplay of cancer cells and TME in association with progression/resistance in patients treated receiving endocrine therapy or in combination with CDK4/6 inhibitors. The authors have done a very nice job in using sound computational approaches to look at the phenotypic interaction and niches and linking it to resistance mechanisms. The paper certainly helps the field to understand the emerging role of myeloid and CD8+ T cells in overcoming resistance to CDK4/6 cell cycle inhibitors. The strength of the manuscript is not only the validation by splitting the data into discovery and validation cohorts but also functionally validating the findings using in vitro experiments. The method section is written very thoroughly. Overall, the dataset and manuscript will be highly valuable to enhance the knowledge in CDK4/6 resistance mechanisms and designing other possible combination approaches. These are some comments and suggestions -

Major Comments:

The computational analysis done to achieve the results are sound, but the interpretations are lacking in the first few result sections. The methods are very well described in the method section but most of the results sections explains the methodology used but does not explain the findings and leaves it up to the reader to read the figures and interpret the findings [line 192-242].

Minor Comments:

1. The authors mentioned performing single cell RNA sequencing on nuclei suspensions, I think it may be clearer to say that single nuclei RNA sequencing was performed since only the tumor nuclei was isolated. Previous papers have shown differences in composition of cells extracted using nuclei vs whole cell sequencing such as adipocytes, mast cells etc are underrepresented using cell suspensions.

2. The author shows the presence of diploid epithelial cells however since these are core biopsies, the presence of normal epithelial cells would be limited, would be good to show that their CNV profile based on inferCNV is flat and they are not really doublets with other cells.

3. Cell type annotation and verification sections seems a bit redundant with the methods. It was a bit unclear as to what results are reported and what observations are made from the Cell-to-cell communication in the "Communication between phenotypically diverse population" section. Would suggest to move the description of the methodology to the methods section and focus on the interpretation of the figures from that analysis.

4. How does one explain the difference in cell type communication between discovery and validation cohort for the same day/tx combination in Figure 3c, between CD8T cells and cancer cells.

5. The clustering shown in the heatmap in Figure 3 A doesn't seem to separate the signals very well, can the authors supplement any additional quality metrics used to identify the 3 clusters/archetypes?

6. The authors haven't looked into the genetic lesions of these patients, do you see a subset of patients driven by some mutation being more resistant?

(Remarks on code availability)

I have reviewed parts of the code, however it is not reproducible right now because of unavailability of data or objects. I

would also recommend having a separate folder that allows to capture the code associated with each figure instead of navigating through multiple files.

Version 1:

Reviewer comments:

Reviewer #1

(Remarks to the Author)

Authors have answered all my concerns

(Remarks on code availability)

Reviewer #3

(Remarks to the Author)

The authors have addressed the raised concerns in a clear fashion and improved the manuscript through the inclusion of further data and explanation.

(Remarks on code availability)

Response to reviewers' comments for: "Breast cancer cells communicate with macrophages to prevent T cell activation during development of cell cycle therapy resistance"

In this document, we address each of the reviewers' comments (italicized) sequentially and respond with a point-by-point description (colored text) of how each comment or question about the manuscript has been addressed.

REVIEWER COMMENTS

Reviewer #1:

Overall Remarks) *"In this study, the authors explore the molecular and cellular crosstalk between cancer cells and immune cells in 2 cohorts (discovery and validation) of ER+PR+ breast cancer patients receiving endocrine therapy (letrozole) alone or combined with CDK4/6 cell cycle inhibition (ribociclib). To this end, authors generated an extensive scRNAseq atlas comprising cancer cells, immune cells and stromal cells of treatment naïve patients, 14 days following treatment and at the end of the treatment (180 days). They use interactome inference to describe how the tumor microenvironment is orchestrated and how it relates to tumor progression and response to treatments. Their analysis show that:*

- (1) resistance to combination treatment is associated with M2 macrophage polarization*
- (2) in treatment resistant tumors, cancer cells increase communication with M2 macrophages*
- (3) response to treatment can be predicted by the immune infiltrated archetype of tumors*
- (4) in responders, effector T cells communicate with M1 macrophages*
- (5) T cell suppression caused by ribociclib is overcome by addition of IL15 in an organoid approach co-culturing cancer cells and T cells*

The study is interesting and provide an important resource for breast cancer research.

However, major concerns need to be addressed to strengthen the current claims"

Response) Our team greatly appreciates these positive comments, and thanks the reviewer for their time, effort and constructive advice.

Comment/Question) - *In Figure 4: Fig4.A: Authors acknowledge the diversity of myeloid cells in the TME but keep a very shallow analysis by only clustering apart monocytes, macrophages and dendritic cells. Authors should push their clustering to highlight how macrophage heterogeneity varies between treatments and archetypes.*

Response) We found this to be an excellent suggestion as higher resolution clustering provided a much richer way to characterize myeloid differentiation and polarization from a monocyte progenitor state to an M1-like or M2-like state. We have used the continuous measure of myeloid polarization to compare myeloid states across treatments and tumor response outcome groups.

We have described our substantially expanded analysis of myeloid cell phenotypic states in the methods section: “Myeloid lineage differentiation structure”.

The first step, as the reviewer recommended, was to push the clustering resolution to uncover the macrophage heterogeneity. We used a Gaussian mixture model and Bayesian Information Criteria to identify the diversity of myeloid phenotype clusters detectable in our single cell dataset (see **Figure S8A** below). We identified 21 distinct myeloid cell states, which we used to characterize the global structure of the underlying myeloid differentiation lineages (including the branching number and locations).

Figure S8A) Visualization of single cell myeloid subpopulations with distinct UMAP phenotypes identified using a Gaussian mixture model, with Bayesian Information Criteria (BIC) used to determine the number of phenotype clusters (n= 21). Cells are colored by cluster. Sample size: n=27127 myeloid cells from 167 of the 173 biopsies from 61/62 patient tumors (Discovery: 10940 cells, Validation: 16187 cells).

Next, we used the slingshot algorithm to identify the global lineage differentiation structure, using slingshot to construct a cluster-based minimum spanning tree (MST) (see **Figure S8B** below). The continuous differentiation trajectory of each lineage was described within slingshot, using principal curves. We incorporated the known biology,

that macrophages differentiate from monocyte progenitors within this analysis. The positioning of cells along each lineage trajectory was then obtained by orthogonal projection onto the curves to provide lineage specific pseudotime trajectories. Average pseudotime values quantified the relative differentiation of each myeloid cell away from the monocyte state.

Figure S8B) Single cell myeloid differentiation trajectory inference. Using clustered myeloid phenotype data in a low-dimensional UMAP space, the slingshot algorithm was applied to construct a minimum spanning tree (MST) between phenotype clusters and determine the number and rough shape of differentiation lineages. The MST was constrained based on the known biology that monocytes progenitors differentiate into macrophage cells. Simultaneous principal curves provided smooth representations of the differentiation trajectories of various lineages (black lines show the 7 trajectories detected). Pseudotime values of each cell (point coloration) were obtained by orthogonal projection onto the curves. This average pseudotime value measures the cell's distance along the MST from the root node (monocyte state). Sample size:: Discovery: 10940 cells, Validation: 16187 cells.

Next, myeloid polarization was measured by the divergence in pseudotime value from the undifferentiated monocyte state (see Figure 4A below). This polarization metric provided a continuous measure of myeloid differentiation from a monocyte state to either an M1- or M2- like state.

Figure 4A) Single cell myeloid polarization from an undifferentiated monocyte state to a differentiated M1-like (blue) or M2-like (red) phenotype. Polarization measured as divergence in pseudotime value from an undifferentiated state (grey; polarization and pseudotime close to zero). Myeloid cells were categorized into a differentiated or undifferentiated state based on their average pseudotime value, using a Gaussian mixture model. A nonlinear division boundary between M1- and M2-like myeloid cells was detected by applying a generalized additive model to the UMAP coordinates of undifferentiated cells. Polarization was then measured by the linear pseudotime divergence of cells from the M1/M2 division boundary. Sample size:: Discovery: 10940 cells, Validation: 16187 cells.

We then compared the average polarization of myeloid cells in biopsy samples of combination ribociclib and letrozole alone treated tumors that were either resistant (growing) or sensitive (shrinking) to treatment. Using a linear model, we detected an increased average M2-like phenotype of myeloid cells in resistant tumors compared to sensitive tumors early (Day 0-14) in combination ribociclib treatment (Est=2.62, sd=0.91, t=2.88, p=0.0051) but not letrozole alone. This result further supported our main conclusion that ribociclib resistant tumors are enriched in M2-like myeloid cells prior to treatment and was consistent with other analysis using established myeloid marker genes to construct knowledge-based characterizations of M2 macrophage gene signatures enrichment (see below).

Comment/Question) - *It is unclear how authors define M1 and M2 macrophages. Authors should use well-defined gene signatures scores for this macrophage states and show how these signatures are enriched by doing GSEA. Such signature can be found in the following studies PMID: 29961579 and PMID: 33545035. In the same lines, authors should provide the transcriptional program defining M1 anti-tumor macrophages.*

Response) As suggested by the reviewer, we have used well-defined gene signatures to show the enrichment of established M2-like macrophage gene sets in the cells identified from our phenotype analyses as having M2-like differentiation. We used state of the art knowledge of cancer-associated myeloid cell markers from recent scRNAseq analyses across human cancer types, obtained from Azizi et al (2018) and Cheng et al. (2021) and enriched this list with markers identified in other recently published studies (listed in Table S30). The knowledge-defined marker gene set was then used to measure gene set enrichment (GSE) using two comparable methods: Gene Set Variation Analysis (GSVA), a GSE method that estimates variation of gene set activity over a sample population in an unsupervised manner and Pathway Level Analysis of Gene Expression (PLAGE), a GSE method that estimates single cell pathway activity using an SVD-based approach. We then examined the relationship between these single cell gene signature scores and the data-derived axes of myeloid differentiation and polarization (Figure S9). We found strong consistency between the different methods to construct gene set signatures. We also found close agreement between the degree of M1-/M2-like differentiation of myeloid cells between the data driven (UMAP/pseudotime reconstruction approaches) and knowledge-driven approached (GSVA and PLAGE). We examined the pairwise correlation between these single cell differentiation metrics and found a strong Pearson correlation coefficient of between 0.35 and 0.93, with a mean of 0.62.

Figure S9) Consistent characterization of myeloid cells into M1-/M2-like states between data-driven analyses (UMAP and pseudotime reconstruction) and knowledge-based characterization approaches (M2 macrophage gene signatures enrichment; Table S30). A. Scatter plots show enrichment of the knowledge-defined M2-like differentiation gene set signatures (coloration) in myeloid cells (points) with polarization along pseudotime trajectories (curves) towards high UMAP 2 differentiation scores (y-axis). Consistent results were obtained when characterizing M2-like differentiation by their expression of established marker genes using Pathway level analysis of gene expression (PLAGE; singular value decomposition) (left-side) or Gene set variation analysis (GSVA) (right side). B. Agreement between myeloid polarization (x-axis) from an M1-like state (negative values) to an M2-like state (positive values), measured by pseudotime reconstruction, and the two approaches to quantify M2-like gene set enrichment scores using PLAGE (y-axis) or GSVA (color). The non-linear association between gene set enrichment and pseudotime polarization (black line) was characterized using a generalized additive model (PLAGE trend: edf=1.99, F=8273, p=2e-16; GSVA trend: edf=2.00, F=2015, p=2e-16). C. Correlation plot showing strong positive correlation (color) and consistency between all metrics to quantify M2-like myeloid differentiation. Circle color indicates the Pearson correlation coefficient between pairs of metrics (red= positive, blue= negative, grey=no correlation) and the size of the circle shows the absolute value of the coefficient (coef:: mean=0.62, range=0.35-0.93). Sample size= 27127 myeloid cells from 62 patients.

Additionally, to define the transcriptional program of M1- and M2-like myeloid cells, we characterized the underlying gene expression changes as cells are polarized along the pseudotime M1/M2 trajectories from the monocyte progenitor state. For each gene, we fitted a general additive model (GAM) to detect the potentially nonlinear relationship between polarization pseudotime and gene expression across myeloid cells. We identified genes that exhibited significant changes in expression along polarization pseudotime and explored the top 40 genes that were linked to increased and decreased polarization. We identified a core transcriptional program defining M1-/M2-like myeloid polarization consisting of 18 known myeloid differentiation markers. We have presented a heatmap visualizing the dynamic changes in each gene with macrophage polarization (Figure S8C).

Figure S8C) Core transcriptional program defining M1/M2 like myeloid polarization. Heatmap shows the single cell expression of 18 genes (rows) that have a known association with M1/M2 differentiation and showed increasing or decreasing expression with polarization along pseudotime trajectories. Columns represent single myeloid cells that are ordered by their polarization pseudotime value (annotation bar above indicates myeloid subpopulation cluster annotation). Genes dynamically expressed across polarization pseudotime were identified using a general additive model to test the potentially nonlinear relationships between gene expression and polarization pseudotime following Van den Berge et al. 2020. M1/M2 differentiation association genes were identified from the 40 dynamically expressed genes that increased and decreased in expression with pseudotime. Sample size:: Discovery: 10940 cells, Validation: 16187 cells.

Finally, we have provided panels of myeloid UMAP phenotype landscape overlays, showing the single cell expression of each of these genes across the landscape of myeloid phenotypic heterogeneity (Figure S10). Together, these complementary approaches have provided consistent results that refine our definition of M1- vs M2-like macrophage cell states.

Figure S10A) Expression of M2 macrophage associated genes across the myeloid cell phenotypic landscape. M2 macrophage marker genes are overlaid onto the UMAP dimension reduction coordinates of single cells from the discovery and validation cohorts (Sample size:: Discovery: 10940 cells, Validation: 16187 cells). Points indicate single cells (color gradient= scales($\log(1+CPM)$) expression). Cells with similar gene expression profiles are located closer together.

Figure S10B) Expression of M1 and dendritic cell (DC) myeloid cell associated genes across the myeloid cell phenotypic landscape. M1/DC macrophage marker genes are overlaid onto the UMAP dimension reduction coordinates of single cells from the discovery and validation cohorts (Sample size:: Discovery: 10940 cells, Validation: 16187 cells). Points indicate single cells (color gradient= scales($\log(1+CPM)$) expression). Cells with similar gene expression profiles are located closer together.

We added a section to the results in which we describe: i) the enriched characterization of myeloid phenotypes, ii) the use of well-defined gene signatures to support the M1/M2-like polarization and iii) definition of the transcriptional profile of these cell states. We end this section by stating:

“Together, these analyses provided multiple complementary lines of evidence supporting our data-driven characterization of the polarization of myeloid cells from a monocyte progenitor to either an M1- or M2-like state being the primary axis of macrophage heterogeneity.”

Comment/Question - *Authors should look and show canonical M1/M2 markers including Arg1, MRC1, CD80, TNF etc...*

Response) We have examined the potentially non-linear association between established M2 myeloid cell markers and our data driven description of single cell differentiation (UMAP and pseudotime polarization). As expected, we found that increased myeloid polarization is linked to higher gene expression across the set of canonical markers (Figure S9D). We provide scatterplots showing this relationship for Mannose receptor C-type 1 (MRC1), Cathepsin B (CTSB), Macrophage Scavenger Receptor 1 (MSR1), Colony Stimulating Factor 1 Receptor (CSF1R) and Macrophage-Associated Antigen (CD163). Other specific markers that the reviewer recommended were also investigated and found to be of generally too low coverage to draw robust inference. We instead provide additional myeloid UMAP overlays for the expression of a set of 18 myeloid differentiation markers found to be associated with myeloid polarization (see Figure S10 above).

Figure S9D) Increased myeloid polarization is linked to higher gene expression across a set of established M2 myeloid cell markers (generalized additive model: non-linear association: $\text{edf}=1.99$, $F=8273$, $p=2e-16$). Scatted plots show the association between myeloid cell polarization, measured by pseudotime reconstruction, and their M2-like macrophage gene set signature (PLAGE enrichment score ($n=36$ genes)), which was derived from pan cancer studies of cancer-associated myeloid cells (Table S30). In each panel, single myeloid cells from the FELINE trial (points; discovery and validation cohort cells) are colored by their gene expression of canonical M2 markers from this signature. The non-linear association between myeloid polarization and M2-like PLAGE signature scores are summarized by a generalized additive model (solid line; shaded regions= 95% confidence interval). Sample size= 27127 myeloid cells from 62 patients.

Comment/Question) - Overall, authors should use state of the art knowledge of the cancer-associated myeloid cell field so readers can integrate these data with the current knowledge. In particular the M1/M2 polarization dogma has been extensively revisited in light of the scRNAseq data performed in various human cancers. Several studies have shown that the functional dichotomy does not necessarily align with the phenotype of M1 and M2 macrophages.

Response) Through the set of additional clustering, pseudotime reconstruction and gene set enrichment analyses described in our earlier responses, we have integrated state-of-the-art knowledge of cancer-associated macrophages, from various published sources (listed in Table S30). This provided multiple complementary lines of evidence in support of our characterization of the relatively continuous polarization of myeloid cells from a monocyte progenitor to either an M1- or M2-like polarization state. Our quantification of polarization was strongly correlated with knowledge-based gene set

enrichment scores derived from the scRNA studies of Azizi et al (2018) and Cheng et al. (2021). The pseudotime analysis, using slingshot, indicates that there is substantial branching in the myeloid phenotype landscape, leading to multiple differentiation trajectories and endpoints. This is consistent with the reviewer's point that the functional dichotomy of macrophages goes further than the divergence between M1-/M2-like cells. However, taken together, all our results point to this dichotomy being the principal axis of transcriptomic heterogeneity amongst macrophages and a consistent trait across tumors that is indicative of ribociclib response prior to/early in treatment (Figure S11).

Figure S11) M2-like macrophage differentiation in growing ribociclib-resistant tumors confirmed using various knowledge- and data-based approaches to measure myeloid phenotypes. Box and whisker plot showing the increased average M2-like phenotype of myeloid cells in resistant tumors (red) compared to sensitive tumors (blue) early (Day 0-14) in combination ribociclib treatment (left panels) but not letrozole alone (right panels). Consistent results were obtained when characterizing M2-like

differentiation by their expression of established marker genes using either Gene set variation analysis (GSVA) or Pathway level analysis of gene expression (PLAGE; singular value decomposition) as well as by characterizing myeloid polarization through pseudotime reconstruction (slingshot applied to UMAP coordinates). Linear model comparisons of each M2-like phenotype metric (rows) made between resistant and sensitive tumors under combination ribociclib (GSVA Est=0.11, sd=0.030, t=3.51, p=0.00078; PLAGE Est=3.05, sd=0.86, t=3.56, p=0.00067; Polarization Est=2.62, sd=0.91, t=2.88, p=0.0051) and letrozole alone treatment (GSVA Est=-0.035, sd=0.039, t=-0.88, p=0.38; PLAGE Est=-2.56, sd=1.32, t=-1.93, p=0.059; Polarization Est=-3.69, sd=1.72, t=-2.15, p=0.037). Sample size: 61 patient tumors (Treatment: 37 combination ribociclib, 24 letrozole alone).

Comment/Question) - *Authors claim that treatment resistant tumors promote M2 differentiation while sensitive tumors promote M1 differentiation. To prove that point, authors should perform a trajectory analysis at time 0, 14 and 180 to confirm the differentiation of myeloid cells towards a M2-like or M1-like profile in resistant or sensitive tumors respectively.*

Response) Following this reviewer's suggestion we assessed the trajectory of mean myeloid cell polarization over time (day 0, 14 and 18). We used the results of the data-driven (UMAP differentiation and pseudotime polarization) and knowledge-based (PLAGE and GSVA) approaches as inputs to generalized additive models that characterize the temporal trajectories of myeloid phenotypic change in ribociclib and letrozole resistant and sensitive tumors (Figure S12). We found that across all myeloid phenotype metrics, the ribociclib resistant tumor had greater M2-like differentiation prior to and early in treatment compared to sensitive cells. As expected, we did not find this difference when comparing letrozole resistant and sensitive tumors. Consistent with the findings in the main text, these results show that a greater abundance of pro-tumor M2-like macrophages pre-exist within tumors and that grows during CDK4/6i treatment.

To explain and clarify this, we state in the results that: "macrophages in growing ribociclib-resistant tumors had greater M2-like differentiation prior to and throughout treatment in both the discovery and validation cohorts, while tumors shrinking during ribociclib treatment had more M1-like macrophages (**Figure 4A, right panel: left subpanels**) (est=0.34, df=32.66, t=3.04, p<0.005). Consistent results were obtained when measuring myeloid phenotypes using knowledge-defined gene set enrichment analysis and when comparing pseudotime polarization (Figure S11; Figure S12)."

Figure S12) Differences in average myeloid phenotype before and during treatment (columns), in tumors resistant (growing=red) or sensitive (shrinking=blue) to therapy. Pathway trends across tumors were determined using a generalized additive model (solid lines). Confidence intervals of model estimates are shown by shaded regions. The mean M2-like phenotype of each tumor sample was measured using four different approaches (rows). The M2-like differentiation was inferred by applying UMAP dimension reduction to myeloid transcriptomic profiles. In contrast, the M2-like GSVA and M2-like PLAGE scores were calculated by measuring the gene set enrichment (GSE) of established myeloid markers using either Gene set variation analysis (GSVA) or Pathway level analysis of gene expression (PLAGE). Finally, M2-like polarization was measured through pseudotime reconstruction (slingshot applied to UMAP coordinates).

Comment/Question) - *In Figure 6: It is unclear why the co-cultures do not include myeloid cells since authors claims these are the main player in the immunosuppression of T cells. In addition, this co-culture experiments could help address the claim made by the authors that treatment resistant tumors drive the differentiation of monocytes toward the M2 phenotype.*

Response) Initial cocultures were performed in the absence of myeloid cells for figure 6 to demonstrate direct effects of IL-15 enhanced activation of CD8+ T cell activity in resistant tumors. To demonstrate the consistency of this effect of IL-15 in combination with ribociclib in the presence of myeloid cells, we have extended our coculture system to model tri-cultures of cancer, macrophage and T cells. We have performed tri-culture experiments of cancer cells (CAMA1), myeloid cells (THP1-M0 like macrophages), and patient derived CD8+ T cells under ribociclib, IL-15 or combination treatment. This experiment confirmed *in vitro* that the combination of ribociclib and IL-15 promotes cancer control via T cells killing in presence of monocyte derived macrophages (see full details in the next-but-one comment/response).

To examine the impact of cancer communication on the differentiation of monocytes toward the M2 phenotype, we have also integrated the well-defined myeloid differentiation gene signatures (described above) with published myeloid sequencing data from controlled cancer-myeloid coculture experiments (using a transwell setting to isolate cell communication effects) (see next comment/response for more details).

Comment/Question) - *In those lines, authors should: 1- Co-culture monocytes isolated from PBMC of healthy donors with ribociclib-sensitive or -resistant spheroids for 7 days and assess the impact of co-culture on the differentiation of monocytes into macrophages and the polarization of macrophages. (To confirm in vitro that ribociclib-resistant tumour cells induce the differentiation and polarization of monocytes into immunosuppressive M2 macrophages)*

Response) Macrophage polarization toward an M2-like phenotype during cancer cell coculture has been demonstrated *in vitro* across various cancer types. We have indicated this in the main text, by stating: “The ability of cancer cells to facilitate polarization of monocytes toward an M2-like phenotype has been previously well characterized *in vitro* under coculture across multiple cancer types including glioblastoma, lung, and breast cancers (Gattas et al 2021, Rebelo et al, 2018, Shi 2019).”

Additionally, we confirmed that within *in vitro* cancer-myeloid cocultures, breast cancer cell communications can induce the predicted form of M2-like myeloid differentiation. We used the published and publicly available molecular dataset of Hollmén *et al.* (2015) (GSE75130) to analyze the impact of cancer cell coculture on known myeloid M2-like marker gene expression (Table S30). They performed controlled monocyte coculture experiments with breast cancer cells and used a transwell experimental setting to isolate the effects of cellular communication. This allowed cancer-macrophage communication crosstalk across a PET membrane but inhibited direct cell contact. They performed whole transcriptome sequencing of human monocytes that were either cocultured with breast cancer cell lines or grown in monoculture.

We obtained this data from GEO and extracted the M2-like marker gene expression profiles (genes listed in Table S30). Linear models we used to perform differential expression analysis of these genes between monocytes cultured alone (monoculture) or with cancer cells (coculture). We used FDR to correct p-values for multiple comparisons. We then identified monocyte genes that were differentially expressed between culture conditions.

We found that the majority of the differentially expressed M2 markers (15/19 genes) were upregulated in cocultured monocytes compared to monoculture conditions (Figure S14). This supported the identified impacts of cancer cell communication on myeloid cell phenotypes. We state that this: “experimentally confirmed the ability of cancer cells to induce myeloid polarization towards an immune-suppressing M2-like phenotype”.

Figure S14) Monocyte cells increase expression of established M2-like marker genes when cocultured with breast cancer cells, compared to monocyte monocultures grown in isolation. Data was generated by Hollmén *et al.* 2015, using a transwell experimental setting to isolate the effects of cancer-myeloid communication on myeloid phenotype from effects of direct contact. They performed

whole transcriptome sequencing of mono-/co-cultured myeloid cells. We used differential expression analysis to compare the expression of known M2-like macrophage marker genes (Table S30) between culture conditions. Heatmap shows the fold change in expression of 19 differentially expressed M2-like macrophage marker genes when monocytes were grown in cancer co-cultures. Most differentially expressed M2-like macrophage markers (rows) were increased in cancer co-cultures (red; 14/19 marker genes), whilst only a few decreased (blue).

References

Hollmén M, Roudnicky F, Karaman S, Detmar M. Characterization of macrophage--cancer cell crosstalk in estrogen receptor positive and triple-negative breast cancer. *Sci Rep* 2015, 5, 9188. PMID: 25776849.

Gattas, M.J., et al. Heterotypic Tridimensional Model to Study the Interaction of Macrophages and Glioblastoma In Vitro. *Int. J. Mol. Sci.* 2021, 22, 5105. PMID: 34065977.

Rebelo, S.P., et al. 3D-3-culture: A tool to unveil macrophage plasticity in the tumour microenvironment, *Biomaterials, Volume 163*, 2018, 185-197, ISSN 0142-9612. PMID: 29477032.

Shi SZ, et al. Recruitment of monocytes and epigenetic silencing of intratumoral CYP7B1 primarily contribute to the accumulation of 27-hydroxycholesterol in breast cancer. *Am J Cancer Res.* 2019, 9(10):2194-2208. PMID: 31720082.

Comment/Question) - 2- Co-culture sensitive or resistant tumour cell spheroids with monocyte derived macrophages and activated T cells +/- Ribociclib +/- IL-15. (To confirm in vitro that ribociclib IL-15 combination can promote T cells spheroid killing in presence of monocyte derived macrophages polarized tumor cells.

Response) Following the reviewer's suggestion, we cocultured cancer cells in spheroids with either monocyte derived macrophages (THP1-M0-like macrophages), CD8+ T cells or in tricultures containing all three cell types. We performed the requested experiment in which ribociclib treatment and IL-15 concentration were varied across treatments and compared cancer growth to DMSO untreated controls.

Cancer cells were cocultured with THP1-M0 like macrophages for seven days prior to T cell addition to allow for macrophage polarization. Cancer cell growth was inhibited under ribociclib treatment compared to control for all coculture and tri-culture conditions relative to control. Additionally, IL-15 enhanced control of cancer cell growth in both cancer + T cell coculture and cancer + macrophage + T cell tri-culture at increasing concentrations of IL-15.

This experiment showed *in vitro* that greater IL-15 concentrations increasingly control cancer growth in T cell or T cell and macrophage cocultures, overcoming the ribociclib immune suppression effects observed at low IL-15 concentrations (Figure S21) (edf=1.27, F=9.50, p=2.46e-5). This confirmed that ribociclib + IL-15 combination

promotes cancer control via T cells killing in presence of monocyte derived macrophages.

We state in the main text that: “We confirmed, using cancer-macrophages-T cell tricultures, that in presence of monocyte derived macrophages the combination of ribociclib and IL-15 promotes cancer control via T cell killing (Figure S21).”.

Figure S21) Impact of IL-15 concentration (x-axis) and ribociclib treatment (color) on the growth rate of cancer cells (CAMA-1 ribociclib sensitive) when cultured alone, or with either patient-derived CD8+ T cells, myeloid cells (THP1-M0-like macrophages) or both (panels and shaped indicate spheroid composition). Cancer population growth rates of replicate spheroids were measured over 12 days (points). Sample size = 96 spheroids, 4 cell type compositions, 4 IL-15 concentrations (0/0.1/0.5/1 ng/mL), 2 ribociclib doses (0/ 1uM), 3 replicates. A generalized additive model characterized the expected cancer growth rate under each composition and treatment (solid lines=predictions; shaded region =95% confidence intervals). IL-15 slowed cancer growth in T cell coculture, but not monoculture, populations (IL-15 effect in T cell cocultures without macrophages: edf=1.92, F=18.14, 2.62e-7). IL-15 also slowed cancer growth in cancer-macrophage cocultures (edf=1.27, F=9.50, p=2.46e-5). Ribociclib decreased the growth rate of cancer populations across coculture conditions (est=-6.51, t=-6.92, p=9.62e-10). The combination of ribociclib and higher dose IL-15 enhanced the control of cancer growth in both cancer-T cell and cancer-macrophage-T cell cocultures (edf=1.89, F=7.80, p=0.00035). Greater IL-15 concentrations increasingly control cancer growth in T cell or T cell and macrophage cocultures,

overcoming the ribociclib immune suppression effects observed at low IL-15 concentrations (indicated by narrowing of gap between ribociclib and control treated cocultures).

Comment/Question) - *T cell activation, survival and exhaustion should be evaluated within the different conditions.*

Response) Following the reviewer's recommendation, we performed additional *in vitro* experiments assessing characteristics of CD8+ T cell growth and activation in monoculture and coculture conditions and across ribociclib and IL-15 treatment conditions. Total ATP was quantified, to measure cell viability, following ribociclib and or IL-15 treatment. Ribociclib suppressed T cell viability in the absence of IL-15, however T cells were able to overcome suppressive effects of ribociclib treatment with increasing concentrations of IL-15 (Figure S17A). The growth kinetics of CD8+ T cell proliferation was monitored over time, by measuring aggregated CD8+ T cell area in spheroid plates over six days under brightfield microscopy (Figure S17B). T cell proliferation slowed by 72hrs under 0 and 0.1ng/mL IL-15. In contrast, T cells remained proliferative through six days of culture with 0.5 and 1ng/mL of IL-15. Lastly, production of IFN gamma was measured by ELISA as a cytokine marker of T cell activation in both monocultures of cancer and CD8+ T cells (Figure S17C). IFN-gamma production was significantly higher in T cell monocultures treated with IL-15 compared to DMSO controls. Additionally, IFN-gamma production was further enhanced in T cell cocultures with cancer cells and was unaffected by ribociclib treatment.

In the results we state: "T cell IFN production, ATP and survival were reduced by ribociclib treatment but more greatly increased by IL-15 cytokine treatment (Figure S17)".

Figure S17) Stimulation of T cell viability, activation and proliferation under IL-15 and combination IL-15 + ribociclib treatment. A) T cell ATP levels increased following IL-15 treatment and overcame the inhibitory effect of ribociclib (observable at low IL-15 doses). Scatterplot showing the impact of IL-15 concentration (x-axis) and ribociclib treatment (color) on T cell ATP levels per unit area of spheroid images (i.e. (total T cell ATP)/(T cell spheroid area)). A linear model describes the log-linear relationship between T cell ATP and square root transformed IL-15 concentrations under ribociclib treated and control conditions (solid line). Shaded regions indicate 95% confidence intervals. IL-15 treatment increased per unit T cell ATP levels (est=2.06, df=44, t=36.57, p=2e-16). Ribociclib opposed this T cell metabolic activation (est=-0.54, df=44, t=-6.74, p=2.76e-8). Sample size: 48 T cell populations; 4 IL-15 levels, 2 ribociclib treatments, 6 replicates. B) ELISA measured IFN- γ cytokine production in spheroids containing monocultured cancer cells (CAMA-1), patient derived CD8+ T-cells, or cocultured cancer cells + CD8 T-Cells after treatment with DMSO, 1 μ M ribociclib, 5ng/mL IL-15, or 1 μ M ribociclib+ 5ng/mL IL-15 for three days (N = 2 samples per treatment). Treatment groups were compared using a two-way ANOVA to test for effects of treatment and composition on log₂ IFN- γ levels. Ribociclib had no significant effect on cancer or T cell IFN- γ production (Cancer: est=-0.074, df=12, t=-0.32, p= 0.75; T cell: est=0.035, df=12, t=0.32, p=0.11). IL-15 increased IFN- γ production of T-cells in monoculture and coculture (monoculture: est=1.59, df=12, t=4.91, p= 0.00036; coculture: est=2.07, df=12, t=6.38, p=3.5e-5). Similarly, the combination of IL-15 and ribociclib led to increased IFN- γ production of T-cells in monoculture and coculture (monoculture: est=1.17, df=12, t=3.60, p= 0.0037; coculture: est=2.13, df=12, t=6.58, p=2.61e-5). C) T Cell proliferation increased during IL-15 treatment overcoming ribociclib inhibition. T cell proliferation was monitored in monoculture in ULA spheroid plates as used for coculture experiments as measured by total aggregated T cell area over time under. T cells were plated and concurrently treated with IL-15 (0.1, 0.5, or 1ng/mL) and/or 1 μ M ribociclib treatment 24hrs prior to initial brightfield imaging with Cytation 5 imager. T cell proliferation was monitored every 24 to 48hrs over six days. Sample size: 120 T cell populations; 4 IL-15 levels, 2 ribociclib treatments, 3 replicates.

Reviewer #2:

Overall Remarks) *“The paper titled “Breast cancer cells communicate with macrophages to prevent T cell activation during development of cell cycle therapy resistance” is a comprehensive and well analyzed effort to understand the interplay of cancer cells and TME in association with progression/resistance in patients treated receiving endocrine therapy or in combination with CDK4/6 inhibitors. The authors have done a very nice job in using sound computational approaches to look at the phenotypic interaction and niches and linking it to resistance mechanisms. The paper certainly helps the field to understand the emerging role of myeloid and CD8+ T cells in overcoming resistance to CDK4/6 cell cycle inhibitors. The strength of the manuscript is not only the validation by splitting the data into discovery and validation cohorts but also functionally validating the findings using in vitro experiments. The method section is written very thoroughly. Overall, the dataset and manuscript will be highly valuable to enhance the knowledge in CDK4/6 resistance mechanisms and designing other possible combination approaches.”*

Response) We are very pleased that this and other reviewers have appreciated the efforts that were taken to generate high quality and valuable clinically relevant and validated data. We appreciate the reviewer’s positive comments about the computational approaches taken to analyze this patient data. We are also happy to see that our approaches to validate results through in patient validation and in vitro functional experiments has been communicated in the manuscript and resonates with reviewers. We agree that this combination of clinical, computational and experimental cancer biology provides valuable insights into the role of the tumor microenvironment, particularly the immune component, in CDK4/6 resistance. It is encouraging to read that the novelty, high value and impact of this work is evident, and that the thoroughness of the analyses is thoroughly detailed.

Comment/Question) - *Major Comments: The computational analysis done to achieve the results are sound, but the interpretations are lacking in the first few result sections. The methods are very well described in the method section but most of the results sections explains the methodology used but does not explain the findings and leaves it up to the reader to read the figures and interpret the findings [line 192-242].*

Response) We thank the reviewer for pointing out the need for greater biological interpretation of results within the first few results sections. We have revised this entire segment, moving any details that are descriptive of methods to the correct section and focusing instead on the description and interpretation of results.

The “Cell type annotation and verification” section (previously lines 192-206) has been condensed and now focuses on explaining the broad non-cancer cell types that were identified, describing the “pronounced copy number amplification” of cancer cells (as questioned in the previous comment) and indicating that cell type annotations were consistent across the discovery and validation cohort.

The section that previously followed (previously lines 207-222) has largely been merged into the methods. A shorten description of the approach to decipher communications between cell populations has been incorporated into the result sections describing tumor-wide communication.

The next section (previously lines 223-250) now covers the compositional similarity of tumors. We describe the finding that immune cell types were highly correlated within tumors and that similarly stromal and endothelial abundances were correlated. We retain a short description of how the compositional similarity of tumors was determined (one sentence) and how to interpret the composition UMAP figure presented (again trimmed to one sentence). We then return to describing the biological findings that tumors fall into distinct archetypes and provide support of this conclusion using alternative clustering methods (see below).

Comment/Question) - Minor Comments:

1. The authors mentioned performing single cell RNA sequencing on nuclei suspensions, I think it may be clearer to say that single nuclei RNA sequencing was performed since only the tumor nuclei was isolated. Previous papers have shown differences in composition of cells extracted using nuclei vs whole cell sequencing such as adipocytes, mast cells etc are underrepresented using cell suspensions.

Response) We understand the reviewers point that single nuclei RNA sequencing would be a valid description and yet we also want to retain the term single cell RNA sequencing (and the abbreviation scRNAseq) as it is more widely used and generally understood description for the method in the field. Also, in our other publications we use this term, and it would be inconsistent and likely confusing to the readership to introduce another way of describing the approach here.

To accommodate both these opinions, we have titled the appropriate methods section “Single nuclei RNA sequencing” and written within that section that: “Single cell RNA-Sequencing (scRNAseq) was performed on single nuclei suspensions (i.e. single nuclei RNA sequencing) using 10X Genomics Chromium platform”.

Comment/Question) - 2. The author shows the presence of diploid epithelial cells however since these are core biopsies, the presence of normal epithelial cells would be limited, would be good to show that their CNV profile based on inferCNV is flat and they are not really doublets with other cells.

Response) Cancer cells can be clearly distinguished from non-cancer cells by performing gene copy number analysis of the scRNAseq data using inferCNV (Figure R1). This has been described in the methods, where we state: “cancer cells were identified by their frequent and pronounced copy number amplification using inferCNV”. We see a clear distinction in the inferCNV profile of cancer cells, compared to the flat profile of non-cancer cells.

Figure R1) Clear copy number alteration of cancer cells but not diploid epithelial, immune or stromal cells. Gene copy number variation (CNV) profile of cancer and neighboring non-cancer cells, from on inferCNV analysis of cells from the discovery cohort of the FELINE trial. Blue color indicates copy number loss and red color indicates copy number gain.

Comment/Question) - 3. *Cell type annotation and verification sections seems a bit redundant with the methods. It was a bit unclear as to what results are reported and what observations are made from the Cell-to-cell communication in the “Communication between phenotypically diverse population” section. Would suggest to move the description of the methodology to the methods section and focus on the interpretation of the figures from that analysis.*

Response) Thank you for your help and advice. To avoid the redundancy identified by the reviewer, we have moved methodological details from the Cell type annotation and verification results section across to the methods and removed repeated information. The condensed “Cell type annotation and verification” results section is now focused on presenting the key resulting outputs of our analyses that are essential to understand how the subsequent findings were reached. We have re-written the results section to describe findings of three key analyses. First, we describe the broad non-cancer cell types that were identified. Then we describe how the cancer cells showed “pronounced copy number amplification” (as questioned in the previous comment). Finally, we briefly describe, with reference to the methods and SI, that cell type annotations were verified using known markers and found to be consistent across cohorts.

We agree with the reviewer’s suggestion to move the details of the “*Communication between phenotypically diverse population*” section into the methods. A shorted description of approach has been incorporated into the subsequent communication-focused result section (after the assessment of tumor composition). This is used to introduce ligand-receptor communication measurement and orientate readers unfamiliar with such approaches.

Comment/Question) - 4. *How does one explain the difference in cell type communication between discovery and validation cohort for the same day/tx combination in Figure 3c, between CD8T cells and cancer cells.*

Response) As data was obtained from primary patient biopsies and the cohort samples were purposefully processed independently (for validation). There were bound to be some distinctions between the discovery and validation results. We have been careful to concentrate on the aspects that validate across cohorts.

Likely biological and technical explanations for disparities in the Cancer-CD8 T cell communication contrasts of resistant and sensitive tumors of the discovery and validation cohort are indicated below by considering signaling from one cell type to another under each treatment separately.

i) CD8 T cells to cancer communication under ribociclib was consistent between the discovery and validation cohort pre-treatment but not post-treatment. This is potentially due to the relative rarity of CD8 T cells in the TME across all tumor samples and the slightly lower frequency of CD8 T cell recovery from the post-treatment biopsies in the validation cohort.

ii) CD8 T cells to cancer communication under letrozole was not significantly different in resistant and sensitive tumors of the discovery and validation cohort pre-treatment. Post treatment there was a significant increase in communication in resistant tumors of the validation cohort, but not the discovery cohort. We noticed that under letrozole the validation cohort yielded more marginally significant effects, whereas in the discovery cohort these differences were found to be non-significant. There was evidently slightly lower variation in communication measurements in the validation cohort. This may have reflected the lessons learned throughout the processing and analysis of the independent discovery cohort samples.

iii) Cancer to CD8 T cells communication under ribociclib was consistently lower in resistant than sensitive tumors across timepoints and cohort.

iv) Cancer to CD8 T cells communication under letrozole was higher in resistant than sensitive tumors across timepoints and cohort, except for pre-treatment in the discovery cohort where no significant difference was detected. A notable difference is the much greater cancer to CD8 T cell communication in resistant tumors post treatment within the discovery but to a much lesser degree in the validation cohort. We speculate that this may be due to a larger number of resistant tumors in the discovery cohort entering into the cancer dominated archetype, in which the few remaining CD8 T cells receive large amounts of pro tumor signaling from the many neighboring cancer cells. However, as the primary focus of the project is on ribociclib resistance, we have not pursued this hypothesis further in this study.

Comment/Question) - 5. *The clustering shown in the heatmap in Figure 3A doesn't seem to separate the signals very well, can the authors supplement any additional quality metrics used to identify the 3 clusters/archetypes?*

Response) We have provided four additional quality metrics to support the clustering presented in figure 3A.

First, we present the results obtained by hierarchically clustering the tumor samples based on their logit transformed composition (Figure S2A). This alternative approach provides qualitatively very similar clusters, as can be seen by the close agreement between the major dendrogram branches and the annotations from our tumor archetype analysis (shown in the "Archetype" sidebar).

Figure S2A) Heatmap of the relative abundance of high-quality cells of each cell type within tumor biopsy samples of the discovery and validation cohorts. Tumors samples (y-axis) and cell types (x-axis) are clustered using hierarchical clustering of logit transformed cell type frequencies. For each sample, annotations (sidebar) are provided for the cohort, the archetype assigned by the tumor archetype analysis, treatment received, Day of treatment and the tumors response. (Sample size= 424,581 annotated cells from 173 biopsy samples of 62 patient tumors at 3 timepoints).

Second, we constructed a pairwise distance matrix to show the similarity or dissimilarity of tumor composition between each tumor sample (Figure S2B). We applied hierarchical clustering to this matrix, to characterize groups of samples with similar compositions. Again, clustering of tumor composition by compositional dissimilarity provided broadly cluster to the groupings that we identified using the tumor archetype analysis.

Figure S2B) Heatmap representing the distance matrix of the logit transformed cell type frequencies within biopsy samples from the FELINE trial. Coloration indicates the Euclidean distance between sample compositions (blue=highest similarity, red=greatest difference). Samples are clustered by pairwise distances using hierarchical clustering (dendrogram). For each sample, annotations (sidebar) are provided for the cohort, the archetype assigned by the tumor archetype analysis, treatment received, Day of treatment and the tumors response. (Sample size= 424,581 annotated cells from 173 biopsy samples of 62 patient tumors at 3 timepoints).

Thirdly, we present the Gaussian mixture model results from the comparison between candidate models that each assume a different number and form of clustering in the data (Figure S2C). The probabilistic support for models hypothesizing that the data has different numbers of clusters was compared to determine the appropriate number of archetypes for the data. BIC model comparison showed most support for the hypothesis that there are three archetypes that could be deduced from the data.

Figure S2C) Bayesian information criterion (BIC) comparison of gaussian mixture models assuming different number of tumor composition archetypes. Higher scores indicate greater probabilistic support for gaussian mixture models with that number of archetypes, given the single cell composition data.

Fourthly, we assessed the reliability with which each sample could be assigned to a cluster (Figure S2D). We show an assessment of the classification uncertainty (the probability of misclassification) of each biopsy sample and associated this to the number of cells sampled from that biopsy sample: a variable not used to train the classification model.

We found that the majority of samples have a low classification uncertainty (typically below 10%) and as would be expected we see that the more cells we sample the more accurately we can assign cells to a tumor archetype cluster. The uncertainty-sample size relationship was almost identical between the two treatment arms, indicating the classifications were robust as long as sufficient cell numbers were present. All samples had a classification uncertainty below 0.5, indicating they could be assigned to one of the archetypes.

Figure S2D) Tumor composition archetype analysis classification uncertainty. For each tumor biopsy sample (point) the archetype classification uncertainty (misclassification probability) was associated with the samples cell count with the expectation that more thoroughly sampled tumors should be classified with less uncertainty. The trend line (solid line; generalized additive model expectation) for each treatment group (color) supports this prediction and indicates that classification uncertainty of the least sampled biopsies was still typically below 10% and decreased rapidly with sampling effort.

Together these analyses indicate that the identified clusters identified are consistent across analysis approaches. The benefit of using the gaussian mixture model approach is its automated way to identify the most parsimonious number of clusters in the data, which in our case there is support for three. Overall, we are pleased by the clustering of tumor samples into consistent and biologically meaningful compositional types. Frequently, clustering approaches will lead to clusters in which separation is driven by technical batch effects. (e.g. distinguishing data from different cohorts). Instead, we find reasonable mixing of samples from different cohorts across the clusters, indicating that technical batch differences are not overwhelming biological signal. Cell types clustered into stromal, immune and cancerous groups. This would be highly unlikely to have occurred by chance. As an example, one can clearly see the correlation of vascular endothelial and pericyte cells, which co-localize to form tumor vasculature.

Comment/Question) - 6. *The authors haven't looked into the genetic lesions of these patients, do you see a subset of patients driven by some mutation being more resistant?*

Response) Our previous published work has examined the genetic mutations of cancer cells in this clinical setting (Griffiths et al. (2021) Nature Cancer). In that study, we conducted whole exome sequencing (WES; mean depth 234x) on pre- and post-treatment tumor biopsies of patients receiving CDK inhibition and endocrine therapy. Matched blood was sequenced in parallel to identify somatic mutations. Several genetic mutations, that are known to be frequent in ER+ breast cancer, were regularly detected, including PIK3CA (46%), TP53 (29%), and MAP3K1 (21%). Oncogenic gene copy number alterations were also frequently identified from the WES data. These included gains in AKT3, CCND1, CCNE2, CDK6, FGFR1 and losses in ESR1, RB1 and TP53. These copy number alterations, with the exception of CDK6, were also more frequent in resistant than sensitive tumors, with most present prior to therapy.

Comment/Question) - *Remarks on code availability: I have reviewed parts of the code, however it is not reproducible right now because of unavailability of data or objects. I would also recommend having a separate folder that allows to capture the code associated with each figure instead of navigating through multiple files.*

Response) We have revised the code repository and source data that accompanies the manuscript to allow reproduction of each of the subpanels of the manuscript's figures.

Source data (named: SourceData_Figure#_BriefTitle.csv) is now provided for each manuscript figure. The input csv file provides the data used in each analysis of the figure. We also provide the output dataset produced by the analysis conducted to generate each subpanel (files in the Outputs subfolder with the suffix "_Output.csv"). Source data has been provided for all main and supplementary figures.

Source code has been refactored and partitioned into separate scripts to perform analyses relating to each subpanel (using Figure input to retrieve subpanel specific output).

Response to reviewers' comments for: "Breast cancer cells communicate with macrophages to prevent T cell activation during development of cell cycle therapy resistance"

We thank the reviewers for once again taking the time to examine our manuscript. We are pleased to see that both reviewers are of the view that we have resolved all the previous issues during the revision process. Below are our responses (colored text) to each of their perspectives (italicized).

REVIEWER COMMENTS

Reviewer #1:

Overall Remarks) *"Authors have answered all my concerns"*

Response) The questions raised helped improve the manuscript and we are pleased to have answered them.

Reviewer #2:

Overall Remarks) *"The authors have addressed the raised concerns in a clear fashion and improved the manuscript through the inclusion of further data and explanation."*

Response) We are pleased that the additional data, analysis and explanation has satisfactorily clarified the reviewers concerns and we agree that their suggestions have helped to strengthen the manuscript.